# Feedback contribution to surface motion perception in the human early visual cortex

Ingo Marquardt[1†]*, Peter De Weerd[1,2†]*, Marian Schneider[1], Omer Faruk Gulban[1], Dimo Ivanov[1], Yawen Wang[1], Kâmil Uludağ[3,4]*

[1]Department of Cognitive Neuroscience, Maastricht Brain Imaging Centre (MBIC) Faculty of Psychology and Neuroscience, Maastricht University, Maastricht, Netherlands; [2]Maastricht Center of Systems Biology (MACSBIO), Faculty of Science & Engineering, Maastricht University, Maastricht, Netherlands; [3]Center for Neuroscience Imaging Research, Institute for Basic Science and Department of Biomedical Engineering, N Center, Sungkyunkwan University, Jangan-gu, Republic of Korea; [4]Techna Institute and Koerner Scientist in MR Imaging, University Health Network, Toronto, Canada

*For correspondence:
ingo.marquardt@posteo.de (IM);
p.deweerd@maastrichtuniversity.
nl (PDW);
Kamil.Uludag@rmp.uhn.ca (KU)

†These authors contributed
equally to this work

Competing interests: The
authors declare that no
competing interests exist.

Reviewing editor: Tobias H
Donner, University Medical
Center Hamburg-Eppendorf,
Germany

**Abstract** Human visual surface perception has neural correlates in early visual cortex, but the role of feedback during surface segmentation in human early visual cortex remains unknown. Feedback projections preferentially enter superficial and deep anatomical layers, which provides a hypothesis for the cortical depth distribution of fMRI activity related to feedback. Using ultra-high field fMRI, we report a depth distribution of activation in line with feedback during the (illusory) perception of surface motion. Our results fit with a signal re-entering in superficial depths of V1, followed by a feedforward sweep of the re-entered information through V2 and V3. The magnitude and sign of the BOLD response strongly depended on the presence of texture in the background, and was additionally modulated by the presence of illusory motion perception compatible with feedback. In summary, the present study demonstrates the potential of depth-resolved fMRI in tackling biomechanical questions on perception.

## Introduction

Historically, vision research has focused on the cortical response to boundaries and edges (e.g. *Albrecht and Hamilton, 1982*; *Hubel and Wiesel, 1968*). Perception, however, requires mechanisms by which areas enclosed by boundaries are 'filled-in'. As surface perception requires spreading or integration of information over a large visual field, these mechanisms have been hypothesized to be localized in high-level visual areas (e.g. *Dennett, 1991*; *Gregory, 1972*; *von der Heydt et al., 2003*), and a role of feedback projections in perceptual filling-in has been suggested (*Devinck and Knoblauch, 2019*). Several studies have indeed claimed that early visual cortex does not contribute to the processing of surfaces (*Cornelissen et al., 2006*; *Friedman et al., 2003*; *Perna et al., 2005*). Nevertheless, a large number of human fMRI studies (*Hsieh and Tse, 2010*; *Kok and de Lange, 2014*; *Mendola et al., 1999*; *Pereverzeva and Murray, 2008*; *Sasaki and Watanabe, 2004*) as well as cat (*Rossi et al., 1996*; *Rossi and Paradiso, 1999*) and monkey electrophysiological recording studies (*De Weerd et al., 1995*; and in *Komatsu et al., 2000*; *Lamme, 1995*; *Lamme, 1999*; reviewed in *Lamme and Roelfsema, 2000*; *Lu and Roe, 2007*; *Roe et al., 2005*; *Zipser et al., 1996*) have demonstrated retinotopic signals in response to the perception of surface brightness, colour, and texture. These surface-related neural signals in early visual cortex have raised the question to what extent they reflect feedback. As feedback projections target predominantly superficial

and deep layers in early visual cortex (*Anderson and Martin, 2009*; *Rockland and Pandya, 1979*; *Rockland and Virga, 1989*), this leads to a clear prediction for activity distributions across cortical depth induced by feedback. In the domain of surface perception, only a handful of neurophysiological studies in animals have successfully tested layer-specific distributions of activity during feedback. Using texture-defined surfaces, two neurophysiological studies in monkeys (*Self et al., 2013*; *van Kerkoerle et al., 2014*) revealed complex temporal patterns engaging both deep and superficial layers. A single human fMRI study using a static surface induced in a Kanizsa display (*Kok et al., 2016*) reported cortical deep layer activity compatible with a role of feedback in surface perception. These experiments align with anatomical data indicating that feedback projections can target both superficial and deep layers. Recent optogenetics studies in mice have moreover confirmed that the correlates of feedback in V1 causally depend on activity in high-level visual cortex (*Schnabel et al., 2018*).

Surface perception is thought to interact tightly with mechanisms of contour reconstruction. A number of computational models of surface perception (*Grossberg, 1987a*; *Grossberg, 1987b*; see also *Keil et al., 2005*) have proposed that diffusion-like spreading in a surface feature system is contained within proper retinotopic bounds by local inhibition delivered by boundary representations. Neurophysiological observations of contour-related responses in V2 (*von der Heydt et al., 1984*) and in V1 (*Grosof et al., 1993*) and surface related responses in V1, V2 and V3 (*De Weerd et al., 1995*; *Huang and Paradiso, 2008*) have emphasized the role of early visual areas in this interaction between surface and contour processing.

Separating responses to edges from responses to the interior of a surface is of utmost importance, as contour responses themselves involve feedback (*Lee and Nguyen, 2001*; *Wokke et al., 2013*), and may show a depth distribution of activity in early visual cortex similar to that elicited by responses to surfaces. In the only depth-specific human fMRI study on surface perception to date, *Kok et al., 2016* presented participants with Kanizsa stimuli containing illusory surfaces and contours. The illusory stimuli caused a response at deep cortical depths in V1, suggesting feedback originating from higher cortical areas. However, due to stimulus design and choice of the region-of-interest (ROI), the feedback related signal could be due to the illusory contour or to the illusory surface, because the ROI could have captured activity related to both.

Research using visual illusions to study the neural correlates of surface perception has predominantly used static surfaces with induced percepts of brightness, colour, or texture, while these features were physically absent in these surfaces. We are aware of only one previous study that measured responses to induced motion of a uniform surface, that is without local changes in retinotopic input (*Akin et al., 2014*). In fMRI studies focusing on motion interpolation, feedback-related responses in V1 were most likely driven by contours rather than surfaces (*Meng et al., 2005*; *Muckli et al., 2005*; *Seghier et al., 2000*), and other motion-related V1 responses may have been driven by local elements in a non-uniform surface (*Muckli et al., 2002*).

By contrast, here we used a stimulus (adapted from *Akin et al., 2014*, see *Table 1* for a detailed comparison of stimulus parameters) that consisted of a centrally fixated, luminance-defined disk, of which a sector was removed. The removed sector was limited to the right hemifield, and rotated clockwise and anticlockwise within the right hemifield, thereby inducing a motion percept of the disk. In the left hemifield, the entire half of the disk was static, remained physically identical, and did not contain local elements inducing the movement percept. Two control conditions that eliminated

**Table 1.** Comparison of stimulus parameters in *Akin et al., 2014* and in the present study.

|  | *Akin et al., 2014* | Present study |
| --- | --- | --- |
| Diameter of stimulus | 15° visual angle | 7.5° visual angle |
| Viewing mode | Central fixation task & passive viewing | Central fixation task |
| Rest block duration | 12 s | 18.7 s, 20.8 s, or 22.9 s |
| Stimulus block duration | 12 s | 10.4 s |
| Stimulus luminance | 503 cd/m2 | 163 cd/m2 |
| Mean background luminance | 189 cd/m2 | 8 cd/m2 |
| Oscillation rate of stimulus | 1.04 Hz | 0.85 Hz |

the illusory motion kept the half of the disk in the left hemifield identical as well. That is, the three stimuli differed in global and local perceptual quality, while being physically identical in the left half of the visual field.

These stimuli, hence, provide several advantages: First, because the motion percept is induced without relying on local elements, an fMRI correlate of surface motion cannot be reduced to merely a modified processing of local elements. Second, because the retinal image of illusory and control stimuli was identical in the left hemifield, and because transcallosal connections are restricted to the vertical meridian in primate early visual cortex (*Clarke and Miklossy, 1990*; *Essen and Zeki, 1978*; *Glickstein and Whitteridge, 1976*; *Wong-Riley, 1974*), any difference between stimulus conditions can be attributed unambiguously to top-down feedback effects. Third, the stimulus was large enough so that contributions to the fMRI signal from the surface were separable from contributions from the contour, enabling any feedback signal to be attributed solely to the surface.

Furthermore, we used ultra-high field (UHF) 7T fMRI to test whether the attribution of motion to a locally static, luminance-defined surface leads to a cortical depth-resolved pattern of activity consistent with feedback processing in early visual cortex. While the tools to perform layer-specific recordings have been available in invasive neurophysiology in animals for decades, the analysis of depth-specific activity in humans has only recently become within reach thanks to UHF fMRI and advances in data analysis (*Guidi et al., 2016*; *Huber et al., 2015*; *Koopmans et al., 2010*; *Koopmans et al., 2011*; *Lawrence et al., 2019*; *Marquardt et al., 2018*; *Olman et al., 2012*; *Polimeni et al., 2010*; *Ress et al., 2007*). Our analysis included not only V1 (as in *Kok et al., 2016*), but was extended to V2 and V3.

Notably, in the non-depth resolved fMRI study that inspired our stimulus design, a smaller BOLD response was reported to the grey figure region than to the textured background, which may have reflected a stronger sensory response driven by the textured than by the homogeneously grey surface. Irrespective of whether the BOLD response to the grey figure was negative or positive, we hypothesized that the illusory perception of surface motion would be associated with enhanced activity in superficial and/or deep layers compared to control conditions, in accordance with a contribution of feedback in early visual cortex.

## Results

In accordance with a previous report using a similar stimulus (see our *Figure 1* and *Akin et al., 2014*), but contrary to what could be the generally expected positive response to a luminance increase, we observed widespread negative signal change in the retinotopic representation of our stimuli in early visual cortex of the right hemisphere. This is illustrated here for the experimental condition inducing the illusory motion percept (*Figure 2*, see also *Figure 1—figure supplement 1*). A band of positive activation with transient peaks at the beginning and end of the stimulation was observed at the cortical representation of the stimulus edge (*Figure 2E,F*). The pattern of negative responses to the surface interior and positive activation at the stimulus edge was similar across stimulus conditions (*Figure 3*; and *Figure 5—figure supplement 1*). Control experiments supported the idea that the negative sign of the response was related to the much stronger response to the texture in the background than to the homogenous grey in the figure (see *Figure 7—figure supplement 1* and *Figure 7—figure supplement 2*).

In the cortical representation of the surface (*Figure 2C*), we found increased activity due to the illusory percept of motion in the experimental condition, compared to the control conditions where this percept was absent. Using a mixed-effects model comparison, we found differential activity among the experimental and control conditions with a magnitude that differed among brain areas V1, V2, and V3, as confirmed by a significant ROI (V1, V2, V3) by condition (motion induction, static control, dynamic control) interaction (likelihood ratio (df): 39.6 (4), p<0.0001). Moreover, cortical depth profiles of the activity increase were significantly different between brain areas (likelihood ratio (df) of model comparison with/without cortical depth by ROI interaction: 30.2 (2), p<0.0001).

*Figure 4—figure supplement 1A* plots the contrasts between experimental stimulus and static control, between experimental stimulus and dynamic control, and among the two control conditions, as a function of depth in areas V1, V2 and V3, after spatial deconvolution. The amount of signal change between the control conditions appeared to differ between areas V2 and V3. To investigate this apparent difference between the control conditions in V2 and V3, we ran a mixed-effects model

**Figure 1.** Stimulus Design. (**A**) A 'Pac-Man' figure rotating about its centre served as the main experimental stimulus. This experimental condition is referred to as 'motion induction stimulus'. (**B**) In the first of two control conditions, the same Pac-Man figure as in (**A**) was presented statically, that is without rotating about its centre. This condition is referred to as 'static control'. (**C**) In the second control condition, a figure consisting of a stationary wedge on its left side, and a smaller, rotating wedge on its right side was presented. In (**A**) and (**C**), the angular position of the 'mouth' and the wedge were modulated sinusoidally, in order to create the impression of a smooth, natural, back and forth movement. Importantly, the motion induction stimulus is perceived to rotate as a whole, whereas the dynamic control stimulus creates the impression of a rotating wedge on the right, and a stationary wedge on the left. At the same time, the retinal image of all three stimuli is identical in the left visual field. All stimuli were presented on a textured random noise background in order to enhance figure-ground segmentation. The stimuli, including the texture background, were adapted from *Akin et al., 2014*. See *Figure 1—figure supplement 1* for a higher-resolution image of the Pac-Man stimulus and the texture background. Videos of the stimuli are available online (https://doi.org/10.5281/zenodo.2583017).

The online version of this article includes the following figure supplement(s) for figure 1:

**Figure supplement 1.** High-resolution image of the 'Pac-Man' stimulus and the texture background.

**Figure supplement 2.** Experimental design.

comparison, but restricted to V2 and V3 and the two control conditions. We found a significant ROI by condition interaction (likelihood ratio (df): 16.7 (1), p<0.0001). In other words, we found a statistically significant difference in the pattern of stimulus-induced activation caused by the control conditions in V2 and V3. Moreover, we found a ROI by depth interaction, showing that the depth profiles differed between ROIs (likelihood ratio (df): 3.9 (1), p=0.0491).

Another relevant outcome of the anatomically restricted analysis was that in V3, the dynamic and stationary controls were not equivalent (mixed effects model comparison, limited to V3 and the two control conditions, testing for an effect of 'condition'; likelihood ratio (df): 35.8 (1), p = <0.0001). As the physical motion in the right hemisphere in the dynamic control condition is better matched to the physical motion of the Pac-Man contours in the experimental stimulus, we considered the dynamic control to be superior over the static control condition. Accordingly, *Figure 4* shows the cortical depth profile of the signal gain corresponding to the induced motion effect for the cortical representation of the stimulus centre, using the difference between motion induction and dynamic control condition. The peak of the apparent motion effect was located at ~25% in V1,~50% in V2, and ~40% in V3, relative to the pial surface (where 100% cortical depth corresponds to the white/grey matter boundary).

The ratio of superficial peak positions in the cortical depth profiles of the condition contrast 'motion induction stimulus' vs. 'dynamic control stimulus' was compared with a chi-squared test. The null hypothesis of no difference in the ratio of superficial peak positions between areas V1, V2, and V3 was rejected (chi-squared (df) = 6.82 (2), p=0.033). As a more specific follow-up, we tested whether the ratio of superficial peaks differed between V1 versus V2 and V3 together. Again, the null hypothesis of no difference was rejected (chi-squared (df) = 6.43 (1), p=0.011). Thus, the ratio of superficial peak positions (in the single subject cortical depth profiles) is significantly higher in striate than in extrastriate cortex (i.e. the peak is closer to the cortical surface in the striate cortex).

For the cortical representation of the stimulus edge, the stimulus conditions also caused differential activation among visual areas (likelihood ratio (df) of linear mixed effects model comparison with/without ROI by condition interaction: 22.8 (4), p<0.0001). However, there was no evidence for

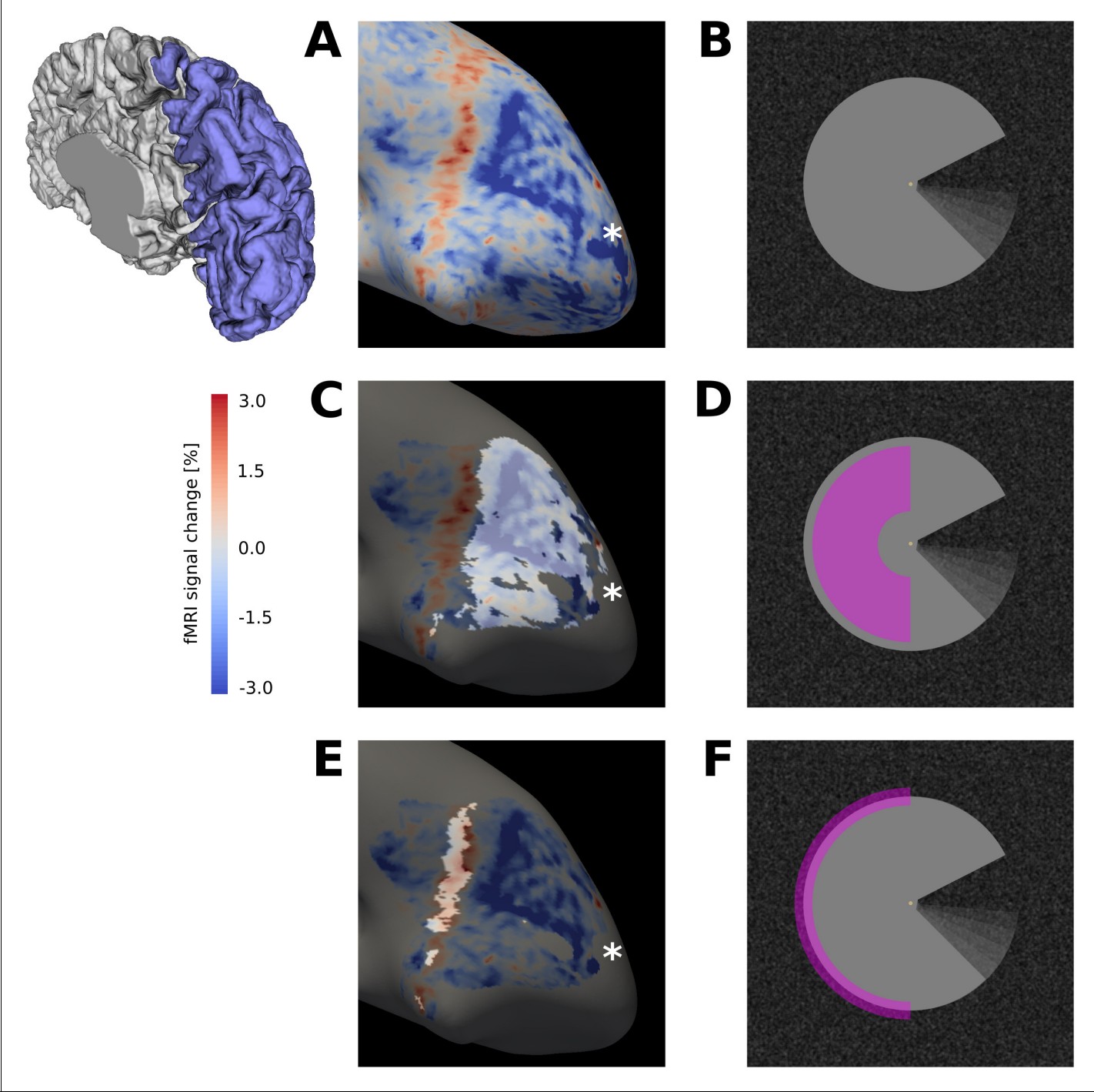

**Figure 2.** Surface activation maps. (**A**) Activation map for motion induction condition (stimulus shown in (**B**)), projected on the inflated cortical surface, for a representative subject (GLM parameter estimates for sustained response). An extended region of negative signal change (blue) is surrounded by a band of positive signal change (red). (**C**) The activation map from (**A**) is masked for V1, and the cortical area that retinotopically corresponds to the centre of the Pac-Man stimulus (**D**) is highlighted. (**E**) Same as (**C**), but the cortical area that contains the retinotopic representation of the edge of the Pac-Man stimulus (**F**) is highlighted. The band of positive signal change corresponds to the retinotopic representation of the edge of the Pac-Man stimulus. The areas highlighted in (**C**) and (**E**) were selected as ROIs for the stimulus centre and edge, respectively. Discontinuities in the ROIs are due to thresholding of the retinotopic map (*R2* >0.15). The asterisk marks the approximate location of the cortical representation of the fovea (**A**, **B**, **C**). The schematic of a right hemisphere next to (**A**) indicates the approximate location of the inflated surface in (**A**, **C**, **E**), highlighted in blue.

The online version of this article includes the following figure supplement(s) for figure 2:

**Figure supplement 1.** The Pac-Man stimulus caused positive and negative fMRI signal changes across visual cortex.

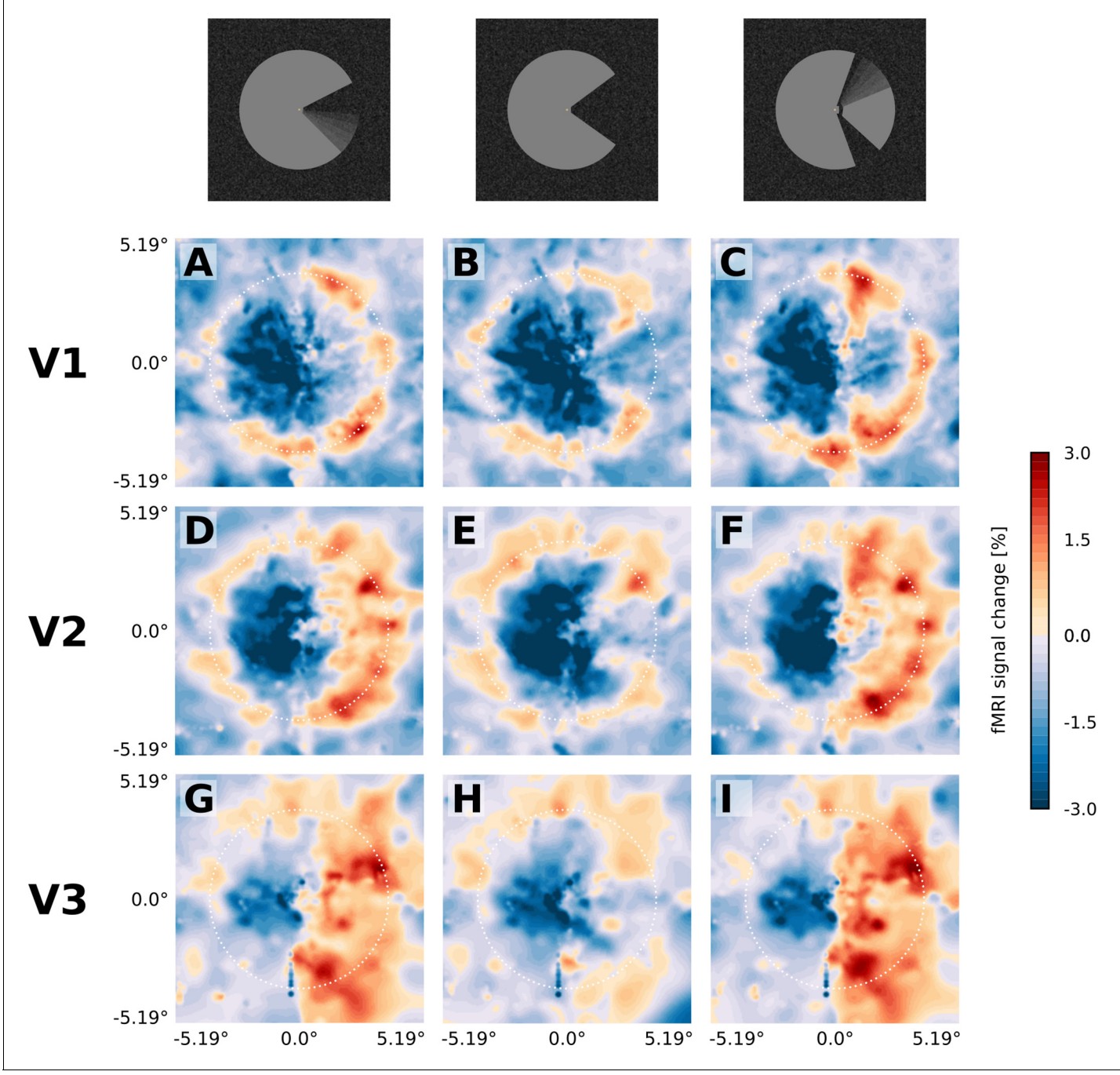

**Figure 3.** Projection of GLM parameters into visual space. The parameter estimates for the three stimulus conditions (motion induction stimulus (A, D, G), static control stimulus (B, E, H), and dynamic control stimulus (C, F, I)) were projected into a model of the visual space based on their retinotopic location, and the size of their respective population receptive fields. The dashed white circles correspond to an eccentricity of 3.75°, that is the radius of the Pac-Man stimulus. In all three stimulus conditions, there is a negative response to the left half of the stimulus. Visual field projections are averaged over depth levels (mean).

differences between stimulus conditions in the cortical depth profiles at the stimulus edge (likelihood ratio (df) of model comparison with/without cortical depth by condition interaction: 1.6 (2), p=0.46); see *Figure 4—figure supplement 2* for cortical depth profiles of apparent motion effect at stimulus edge). This is likely due to the strong feedforward drive due to local contrast at the figure's edge, which may engage neurons about equally across cortical depth.

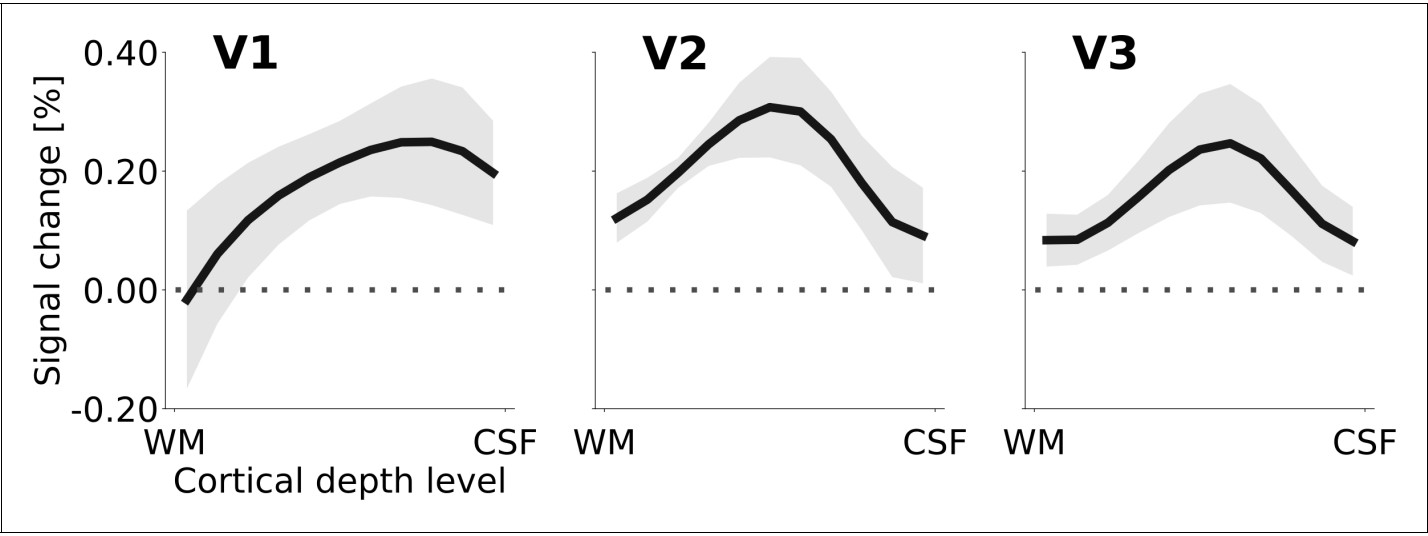

**Figure 4.** Cortical depth profiles of the apparent motion effect for the cortical representation of the stimulus centre (see *Figure 2C & D*). The apparent motion effect was defined as the relative signal change associated with the condition contrast 'motion induction' (*Figure 1A*) minus 'dynamic control' (*Figure 1C*). Shading represents the standard error of the mean (across subjects). See *Figure 4—figure supplement 1* for the same results for all experimental conditions, *Figure 4—figure supplement 3* for single subject data, and *Figure 4—figure supplement 2* for the cortical depth profile of the apparent motion effect at the representation of the stimulus edge.

The online version of this article includes the following figure supplement(s) for figure 4:

**Figure supplement 1.** Cortical depth profiles of all three condition contrasts, with (**A, B, C**) and without (**D, E, F**) spatial deconvolution for removal of signal spread due to draining veins.

**Figure supplement 2.** Cortical depth profiles of the apparent motion effect for the cortical representation of the stimulus edge (see *Figure 2C and F*).

**Figure supplement 3.** Cortical depth profiles of the apparent motion effect for the cortical representation of the stimulus centre.

## Temporal response pattern

In areas V1, V2, and V3, the central region of interest for all conditions exhibited a sustained negative response, whereas the edge region responded with a transient positive signal change at stimulus onset and offset (*Figure 5*). Separately for the sustained and transient responses, we determined response onset time as the first time point at which the signal was significantly different from zero (one-sample t-test, p<0.05, Bonferroni corrected). Interestingly, this revealed that the onset of the transient response at the cortical representation of the stimulus edge preceded the onset of the sustained response in the surface representation by one MRI acquisition time point (i.e. ~2 s; *Figure 5*). The pattern of positive transient and negative sustained responses at the stimulus edge and centre, respectively, was consistent across areas, conditions (*Figure 5—figure supplement 1*) and subjects (*Figure 5—figure supplement 2*). An additional control experiment was performed to investigate whether the temporal dynamics of the responses were similar for a longer stimulus duration (*Figure 5—figure supplement 3*). The results indicate that this was indeed the case, and that the negative response to the centre of the PacMan surface was sustained over long stimulus durations (25 s, compared to ~10 s in the main experiment). Separate cortical depth profiles of the early and late response phases show no evidence for temporal differences in the laminar activation profile (*Figure 5—figure supplement 4*).

Similar to the main experiment (*Figure 5*), a trend towards an onset time difference was also observed in the control experiment (*Figure 6*). In this case, the onset difference is found between stimulus conditions, at the same retinotopic location. However, this onset time difference is not statistically significant, probably due to the small sample size of the control experiment (*n* = 2).

## Spatial response pattern

The spatial distribution of positive and negative signal change is directly visible in the visual field projections (*Figure 3*). As expected for a moving stimulus, the dynamic parts of the stimulus (i.e. the rotating 'mouth' of the Pac-Man, and the rotating wedge of the dynamic control stimulus) caused a

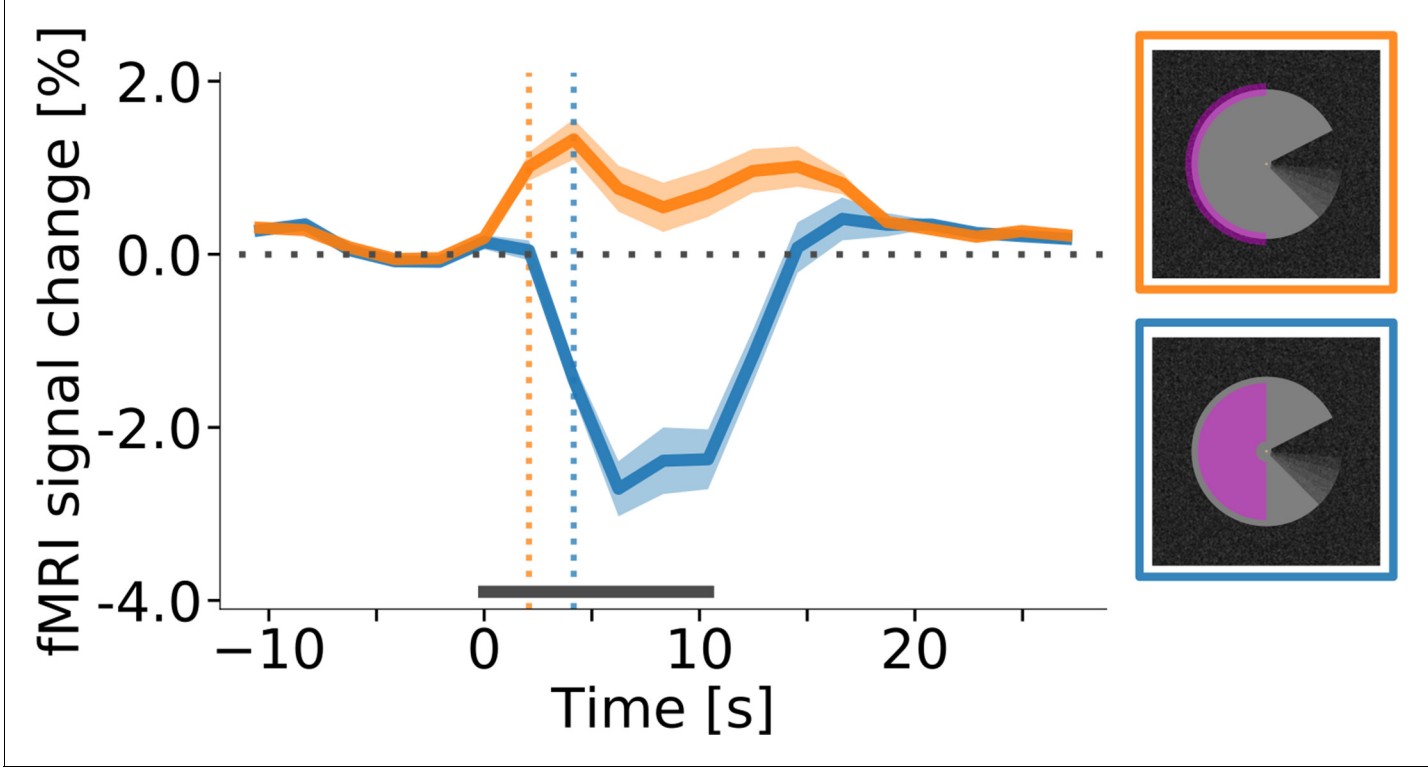

**Figure 5.** Response onset times in V1. (**A**) Event-related fMRI timecourses for regions of interest corresponding to the stimulus centre (blue line) and the edge of the stimulus (orange line). The dotted vertical lines indicate the response onset, defined as the first time point at which the signal was significantly different from zero (one-sample t-test, p<0.05, Bonferroni corrected). The positive response at the stimulus edge precedes the negative response at the stimulus centre by one volume (i.e. by about 2 s), suggesting that the negative response is not caused by the onset of the stimulus, but by its prolonged presentation. The response is shown for area V1 of the right hemisphere, averaged (mean) over subjects, stimulus conditions, and cortical depth levels. The horizontal grey bar marks the duration of the stimulus block. Error shading represents the standard error of the mean (across subjects). (See *Figure 5—figure supplement 1* for same results separately for all areas and conditions, and *Figure 5—figure supplement 2* for single-subject data.).

The online version of this article includes the following figure supplement(s) for figure 5:

**Figure supplement 1.** Event-related fMRI timecourses for region of interest corresponding to the stimulus centre (**A, B, C**) and the edge of the stimulus (**D, E, F**) in the right hemisphere.

**Figure supplement 2.** Single-subject response onset times in V1.

**Figure supplement 3.** Event-related fMRI timecourses from an additional run with longer stimulus blocks, for V1 in the right hemisphere.

**Figure supplement 4.** Cortical depth profiles of the apparent motion effect for the cortical representation of the stimulus centre.

**Figure supplement 5.** Simulation of negative fMRI response at the cortical representation of the stimulus centre.

positive signal change in their cortical representations in V1, V2, and V3 (*Figure 3A,C,D,F,G,I*). All three stimuli caused a negative signal change in the surface's representation in the right hemisphere in V1, V2 and V3 (*Figure 3A–I*). The band of positive signal change seen on the inflated brain (*Figure 2E*) is also apparent in the visual field projections (particularly in *Figure 3D,E,F*). Especially for the static control stimulus, the shape of the stimulus is visible in the visual field projections (*Figure 3B E*), evidence for a high accuracy of the visual field projections across the subjects. The spatial extent of the negative signal change was similar across conditions, but differed across regions; from V1 over V2 to V3, the visual field projections are more blurred, likely due to the increasing neuronal receptive field size in higher-order areas (*Gattass et al., 1981*).

## Background dependence of the negative response

A control experiment was conducted to investigate the effect of the background and of the stimulus shape on the processing of a surface stimulus. The results revealed that the directionality and temporal course of the response is heavily affected by the type of background, but not by the shape of

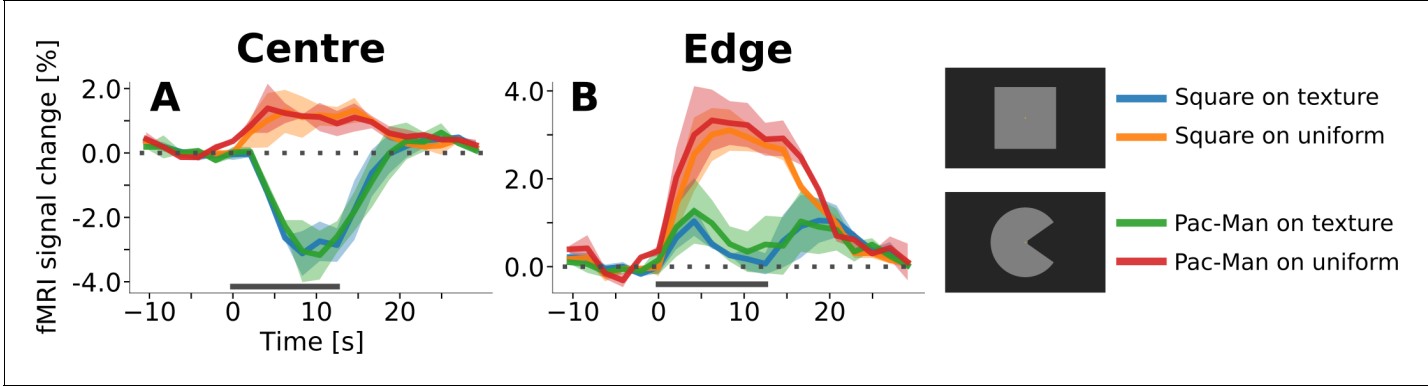

**Figure 6.** Event-related time courses from control experiment with texture background and uniform background, separately for regions of interest corresponding to the retinotopic representation of the centre of the stimulus (**A**) and to its edges (**B**). Irrespective of the shape of the stimulus (square or 'Pac-Man'), there is a positive response to the centre of the stimulus when the background is uniform (A, red and orange lines), and a negative response when the stimuli are presented on a random texture pattern (A, green and blue lines). Interestingly, the positive response has a shorter latency than the negative response. The response to the edges of the stimuli is positive under all conditions (**B**). However, the response amplitude is much stronger when the stimuli are presented on a uniform background. Moreover, the temporal dynamics changes as a function of the background; the response is sustained when the background is uniform (B, orange and red lines), but transient for the texture background (B, green and blue lines). The horizontal grey bar marks the duration of the stimulus.

the stimulus. A negative surface response was only observed when the stimuli were presented on a texture background, irrespective of the stimulus shape (*Figure 7B D*). When presented on a

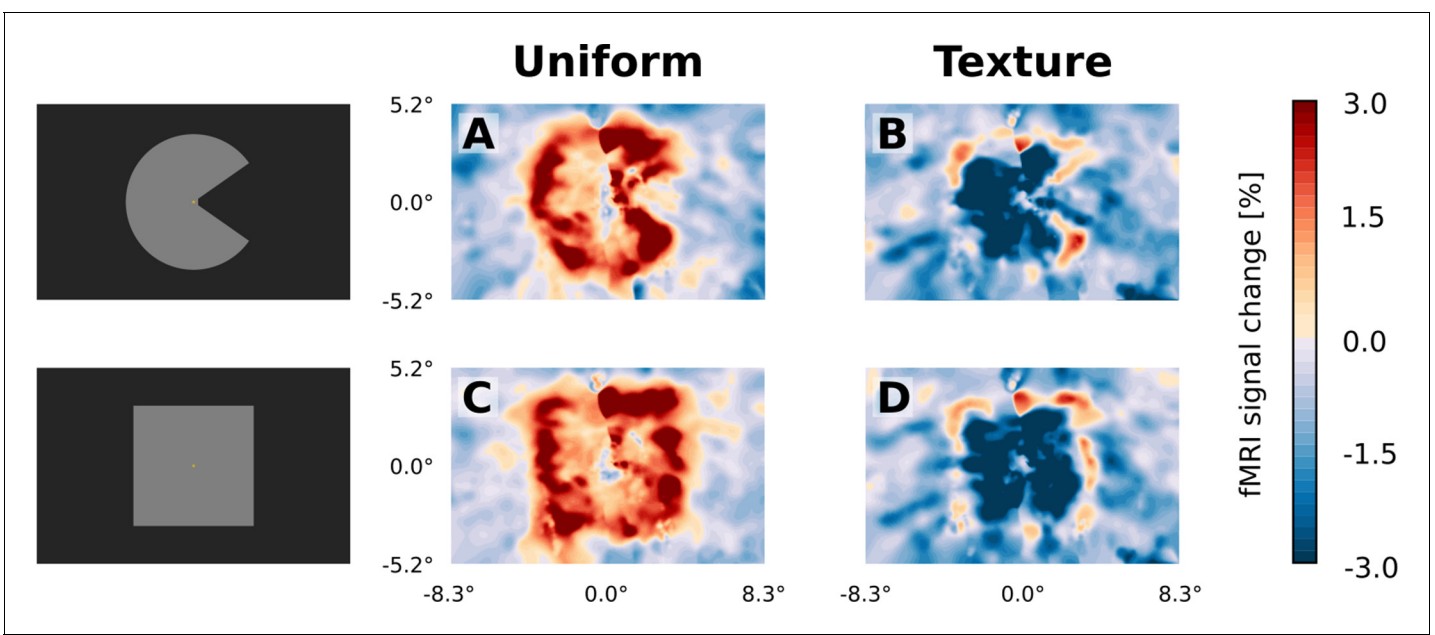

**Figure 7.** Visual field projections of GLM parameter estimates from control experiment with texture background and uniform background, for V1. A 'Pac-Man' figure and a square were presented either on a uniform background (**A and C**) or on a random texture background (**B and D**). When presented on a uniform background, the stimuli caused a positive response, especially at the retinotopic representation of the edges (**A and C**). In stark contrast, the response to the interior of the stimuli was negative when presented on a random texture background (**B and D**). At the edges of the stimuli, a small band of positive activity can still be observed (**B and D**).

The online version of this article includes the following figure supplement(s) for figure 7:

**Figure supplement 1.** Control experiment to investigate the response to the texture pattern, in the absence of any additional stimulus.

**Figure supplement 2.** Additional control experiment, investigating the effect of a texture background on stimulus-induced responses relative to a uniform baseline (n = 1).

**Figure supplement 3.** Stimulus design of the control experiment.

homogenous background, as luminance stimuli are usually presented, the interior of the surface and its edges evoked a positive response (*Figure 7A C*). An additional control experiment showed that the full-screen texture pattern evokes a positive response when contrasted against a uniform rest condition (*Figure 7—figure supplement 1*). Thus, it is safe to assume that the negative surface response was caused by the texture background.

The temporal dynamics of the response in the texture background condition (*Figure 6*, green and blue lines) closely resembled the results from the main experiment (*Figure 5*); showing a transient positive response at the edges and a sustained, delayed, negative response at the surface interior. In contrast, the response to both the interior of the surface and to its edges was positive and sustained in case of a uniform background (*Figure 6*, red and orange lines). Interestingly, these results imply that the temporal shape of the edge response changed as a function of the background condition; in other words, whether the edge response is sustained or transient depends on whether the stimuli are presented on a texture pattern or on a uniform background.

Because the delay in the negative response is observed under all three stimulus conditions in the main experiment (*Figure 5*, *Figure 5—figure supplement 1*), the delay is unlikely to be related to the (apparent) motion of the stimuli. The delay may be due to a slow decrease in activation, following an elevated baseline caused by the texture background. However, we have also observed delayed negative BOLD responses in V1 and V2 under very different stimulus conditions (without texture background), even in the absence of any change in local retinotopic input (see *Marquardt et al., 2018*, p. 176, Figure 5.7 E and F, and p. 177, Figure 5.8, https://doi.org/10.26481/dis.20190829im). Moreover, we have at present no explanation for the temporal dynamics of the response at the stimulus edge (*Figure 6B*). Thus, it is possible that varying physiological contributions to the negative BOLD response contribute to the observed temporal dynamics, and more research will be needed to clarify the origin of these delays.

## Discussion

We have studied neural correlates of perceived surface motion induced in a locally static grey surface on a dark, textured background (*Figure 1*). The motion percept was caused by local edge movement in the contralateral hemifield and spread over the entire surface in the ipsilateral hemifield. We report three main findings: First, the induced percept of surface motion was associated with an fMRI signal increase in the representation of the surface in areas V1, V2 and V3 (*Figure 4*). As the enhanced signal was measured far away from the location where the perceived motion was induced, this signal likely derives from feedback. In addition, the differences in the cortical depth distribution of motion-percept related signal gain among visual areas also supported a feedback origin. Second, we found that the response to the edge preceded the response to the surface by approximately 2 s (*Figure 5*). Third, we observed a negative BOLD signal in the figure representation (*Figure 3*), which depended on the presence of a textured background and was eliminated when the background texture was removed (*Figure 7*). Hence, the signal gain due to the motion percept represented an increase in signal from a negative BOLD signal in the control condition to a less negative BOLD signal in the illusory movement-condition.

### Top-down feedback

The main and control stimuli were 'physically' identical in the left visual field, while the global perceptual quality of the stimulus depended on the right half of the stimuli (*Figure 1*; videos of the stimuli are available online: https://doi.org/10.5281/zenodo.2583017). This stimulus design offers three advantages: First, the surface itself was homogenously grey and did not contain local moving elements, thereby avoiding the interpretative question whether enhanced fMRI activity during surface perception reflects enhanced processing of local elements or an integrated surface motion percept. Second, any changes in activity correlating with a perceptual change from static to moving in the left hemifield were induced by the right hemifield. Anatomical investigations have shown that direct, transcallosal, interhemispheric connections are restricted to the proximity of the vertical meridian in primate early visual cortex (*Clarke and Miklossy, 1990*; *Essen and Zeki, 1978*; *Glickstein and Whitteridge, 1976*; *Wong-Riley, 1974*). This, combined with the fact that the surface motion percept in the left hemifield was induced in the absence of physical changes to the left-hemifield stimulus, renders top-down feedback from higher areas, rather than within-area horizontal interactions, the most

plausible source of the motion percept and associated depth distributions of activity. Third, the cortical region that retinotopically represents the physically constant left side of the stimulus and the one which induces the motion percept (i.e. the 'mouth' of the Pac-Man) were far apart. Thus, it is very unlikely that imprecisions in the retinotopic maps could confound our results. By the same token, the size of our stimulus enabled us to separate responses to the surface from responses to the contours.

Although there was considerable variability among subjects, overall, the cortical depth profiles of the enhanced response due to the illusory motion effect in V1, V2, and V3 suggests that top-down signals may have re-entered at superficial layers in V1, where most of the signal gain due to motion perception was concentrated (*Figure 4*). Re-entrant connections via superficial V1 have been reported in neurophysiological (*McManus et al., 2011*), anatomical (*Martinez-Conde et al., 1999*), and high-field fMRI studies (*Lawrence et al., 2019*; *Muckli et al., 2015*). This re-entrant information may have propagated to V2 and V3 via feedforward pathways, in line with anatomical evidence that the strongest forward projections from V1 to V2 originate in superficial V1 layers 3B and 4B, and arrive across the full extent of layer four in V2 (*Douglas and Martin, 2004*; *Felleman and Van Essen, 1991*). Furthermore, forward projections originating in superficial V1 layers and superficial V2 layers also target layer four in V3 (*Rockland and Pandya, 1979*; *Van Essen et al., 1986*). This pattern of forward projections may explain the activity peak at intermediate depths of areas V2 and V3 (*Figure 8A*). Therefore, although our data do not permit a direct test of the directionality and precise temporal dynamics of information flow, re-entrant feedback at the level of V1 is a plausible interpretation of the present results.

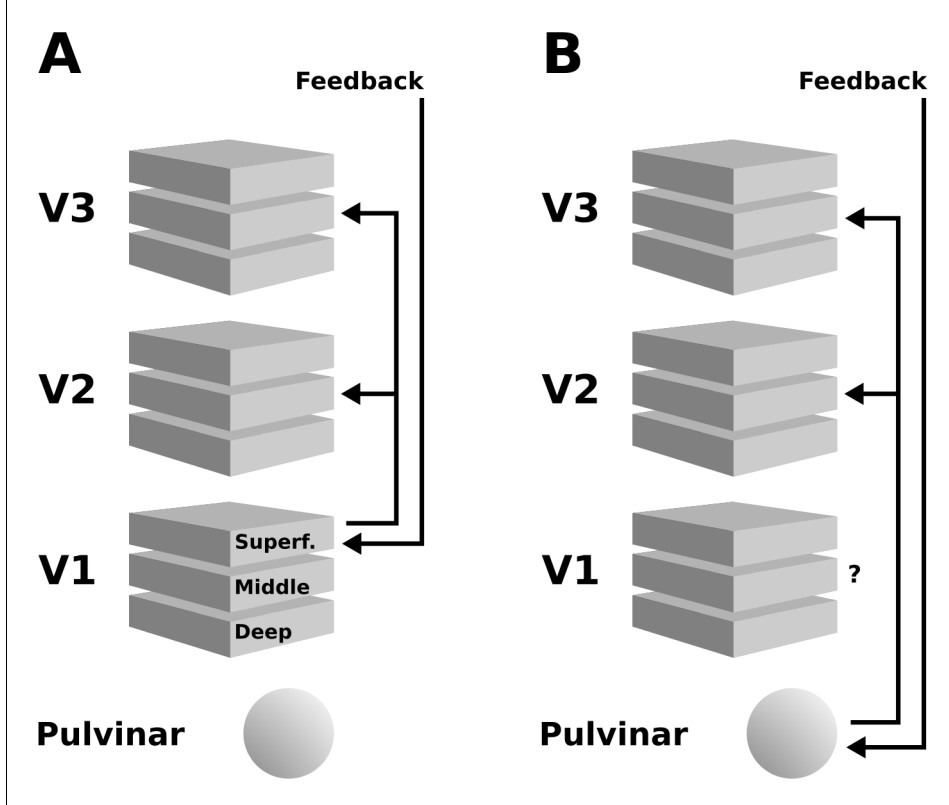

**Figure 8.** Schematic illustration of two possible interpretations of the present results. (**A**) Higher cortical areas may integrate the global motion percept across hemispheres, and send feedback projections to superficial layers of V1. Subsequently, this re-entrant feedback would be sent to V2 and V3 via feedforward connections. (**B**) Alternatively, the pulvinar may act as a 'higher-order relay', and send feedback from higher cortical areas to V2 and V3. These scenarios are not mutually exclusive, and other possibilities exist, such as an involvement of the LGN – see discussion section for details.

An additional contribution to the depth-pattern of activity observed in extrastriate areas may have originated from the pulvinar, the LGN, and possibly other subcortical structures (*Standage and Benevento, 1983*; *Trojanowski and Jacobson, 1977*). The middle layers of extrastriate cortex are the target of projections from the pulvinar (*Benevento and Rezak, 1975*; *Figure 8B*; *Benevento and Rezak, 1976*; *Ogren and Hendrickson, 1977*; *Rezak and Benevento, 1979*), a structure that is sometimes referred to as a 'higher-order relay' because of its role in cortico-cortical interaction (*Sherman and Guillery, 2002*). The pulvinar has been shown to regulate cortico-cortical communication in the visual system based on attentional demands (*Saalmann et al., 2012*). Experiments in humans (*Villeneuve et al., 2005*; *Villeneuve et al., 2012*) and cats *Merabet et al., 1998* have demonstrated a role of the pulvinar in higher-order motion processing (i.e. coherent motion of entire objects, as opposed to local motion). In line with this, *Shimono et al., 2012* have found evidence for an involvement of the pulvinar in the interhemispheric integration of motion information (2012). Moreover, an involvement of the LGN in the perception of illusory motion has been observed by *Akin et al., 2014*, using an experimental design very similar to ours. As the V1 cortical depth profile we observed suggests similar levels of activity in mid-level to superficial layers (*Figure 4*), it is possible that a feedback signal assigning motion to the grey surface re-entered the LGN, was fed-forward to V1, and from V1 to V2 and V3. In summary, both cortical and subcortical sources of re-entrant feedback in lower-level visual areas may have contributed to the observed depth-resolved responses (see *Figure 8*).

The increase BOLD contribution associated with the illusory percept of surface motion is in line with other fMRI studies for a range of surface illusions (*Hsieh and Tse, 2010*; *Kok and de Lange, 2014*; *Mendola et al., 1999*; *Pereverzeva and Murray, 2008*; *Sasaki and Watanabe, 2004*). Compared to *Kok et al., 2016*, who reported a fMRI response enhancement limited to the deepest cortical layers during the percept of an illusory Kansiza triangle, the signal gain we found was focused on superficial to middle layer compartments. Our results resemble somewhat more the superficial activity reported in *Muckli et al., 2015* in response to the completion of occluded visual scenes, and in *Lawrence et al., 2019* associated with feature-based attention. These differences in activity depth profiles could reflect fundamental differences in feedback mechanisms engaged in the stimulus paradigms in the different studies, which is a possibility that should be investigated further. Irrespective of the differences in observed activity profiles over depth, they all support re-entrant feedback signals, which is in line with mounting evidence that, even for the simplest displays, feedback from the highest level of the visual system plays a role (*Lamme and Roelfsema, 2000*; *McManus et al., 2011*; *Roelfsema et al., 2002*; *Schnabel et al., 2018*).

The amplitude of the apparent motion effect (*Figure 4*) is small, compared with the overall response strength (e.g. *Figure 5*). This result is consistent with previous studies, finding small effect sizes of top-down effects relative to bottom-up effects. For example, *Kok et al., 2016* reported a top-down modulation of about 0.12% (their Figure 2A). In our data, the apparent motion effect is strongest at mid-cortical depths in V2, where it reaches an amplitude of slightly more than 0.3%. In previous studies, top-down effects were observed in experiments that contrasted stimulus conditions evoking a positive response, whereas in our case the stimulus induced responses were negative compared to the baseline. We see no principled objection against the interpretability of a condition contrast among control and experimental responses that both yield BOLD responses smaller than the BOLD baseline response.

## Hemispheric imbalances in stimulation

Our stimuli used a stimulus manipulation in the right hemifield, in order to induce an illusion in the left hemifield. This raises the possibility of interhemispheric interactions that may cause a stronger negative BOLD signal in the illusory condition compared to the control condition. In principle, a combination of transcallosal, interhemispheric connections (*Clarke and Miklossy, 1990*; *Essen and Zeki, 1978*; *Glickstein and Whitteridge, 1976*; *Houzel and Milleret, 1999*; *Van Essen et al., 1986*; *Wong-Riley, 1974*) as well as feedback of lateral interactions taking place in higher-order visual cortex could affect the interhemispheric balance of activity in V1, V2 and V3. These higher level interactions may also include attentional imbalances. Several studies have demonstrated that spatial attention causes both a positive response in the cortical representation of attended locations, and a negative response at unattended locations (*Bressler et al., 2013*; *Müller and Kleinschmidt, 2004*; *Silver et al., 2007*; *Slotnick et al., 2003*; *Somers et al., 1999*; *Tootell et al., 1998*).

As we indicated above, we propose that the primary cause of the negative BOLD in the grey surface is primarily related to the absence of texture inside the figure compared to the texture baseline. Nevertheless, we verified whether an imbalance between the constant area of the stimulus in one hemifield (where the illusion is probed), and the part of the stimulus in the other hemifield (where the illusion is either generated or prevented in the control conditions) could explain the pattern of results we obtained.

Our data directly speak against this possibility. The right field stimulation in the motion illusion stimulus (*Figure 1—figure supplement 2A*) and the dynamic control stimulus (*Figure 1—figure supplement 2C*) were designed to be similarly high, whereas the right field stimulation in the static control condition (*Figure 1—figure supplement 2B*) was designed to be low. If an activity imbalance between hemifields had been the primary cause of our pattern of results, we would have expected that the dynamic control condition and the illusion condition would both have induced a similar decrease in activity in the left hemifield representation compared to the static control condition. In addition, in the left hemifield's representation, an increased (less negative) BOLD would have been observed for the dynamic than for the static control condition. Instead, we find that in V1 and V2, the static and dynamic control conditions yield the same activity levels in the left hemifield representation, and that in the left hemifield representation the motion illusion condition gives the *highest* (least negative) response, that is the illusory motion condition yields activity that exceeds (i.e. is less negative than) the activity in both control conditions (see *Figure 4—figure supplement 1*). This shows that the dynamic stimulation in the right hemifield in itself (and a potential associated imbalance in attention) is unlikely to be responsible for the observed pattern of results in V1 and V2.

Note that in V3, we found that the response in the representation of the left part of the stimulus (where the illusion is probed) was smaller for the dynamic control condition than for the static control condition, which suggests that V3, in contrast to V1 and V2, was sensitive to differences in activity between hemifields. Accordingly, when contrasting the motion illusion condition against the dynamic control condition, the positive difference is smaller than when contrasting the motion illusion condition against the static control condition. This shows that, in V3, the former rather than the latter statistical contrast provides the best estimation of the motion illusion, and that overall the dynamic control condition was superior over the static control condition.

## Attention

Although our data do not support that an imbalance in stimulation or attention drove the observed pattern of results, the fixation task we used likely permitted at least a minimal level of attention to the visual field as a whole. The observed depth-resolved data pattern in V1, V2 and V3 therefore raises an interesting set of interrelated questions. Would the observed depth-resolved data pattern reflect computations specifically related to the assignment of motion to a grey region where physical evidence for that percept is lacking, with a potential contribution of attention boosting these computations? Alternatively, would the observed depth-resolved pattern be dominated by a general form of attention allocated to a segmented surface? To address these questions, a possible approach would be to re-run the present experiment four times in a 2 × 2 design, and manipulate the surface feature being induced (e.g., color vs. motion) and the amount of attention available (e.g., by using a hard versus easy fixation task). If the two surface features would elicit different configurations of deep and superficial activity in striate and/or extrastriate areas in the low-attention condition, which each would be strengthened by attention, this would support the idea of depth-resolved activity specifically due to differentiable surface-related computations. If, on the other hand, there would be no activity in the low-attention condition to either of the two induced surface signals, and making more attention available would reveal the same depth-resolved patterns of activity irrespective of the induced surface feature, this would support the interpretation of a depth-resolved signal primarily driven by a general attention process. This approach is similar to how *Lawrence et al., 2019* distinguished effects of contrast from effects of attention in a depth resolved fMRI study in early visual cortex. We have not done these experiments, and therefore we cannot ascertain how strongly the patterns of depth resolved activity in our study reflect operations specific to the induced surface feature.

The data in *Lawrence et al., 2019*, however, may provide an element suggesting at least some level of specificity in our data. *Lawrence et al., 2019* manipulated attention of their participants to a moving surface induced in a plaid stimulus with luminance-defined components. They found a bias

of activity towards superficial layers during attentional feedback that was the same for V1, V2 and V3. *Lawrence et al., 2019* interpreted this as evidence for a feature-based attentional feedback signal. If this is indeed a depth-dependent signal that reflects the general capacity of feature-based attention, attention to the induced motion in our study would be expected to yield a depth resolved signal similar to that in *Lawrence et al., 2019*. Instead, in the present study, we found a bias towards activity in superficial layers in V1, with a significantly stronger bias for activity in middle and deeper layers in V2 and V3. This suggests that the feedback processes due to the attentional manipulation in *Lawrence et al., 2019* and due to the motion induction in the present study differ, and that the likely attentional contributions in our study are at least to some extent specific to the computations underlying motion induction. We cannot fully exclude that our observations reflect a general attentional contribution rather than computations specifically related to the perception of surface motion. These are exciting open questions, and it is encouraging that solving these questions is now within reach of high-field fMRI.

## Edge responses preceding surface responses

Psychophysical experiments (*Paradiso and Nakayama, 1991*) and neurophysiological experiments (*Huang and Paradiso, 2008*) have suggested that surface brightness may fill in from the edge over a time period of ~100 ms, depending on the size of the surface. This interpretation of the reported data is in line with computational models that propose a primary analysis of the visual scene to delineate contours, followed by a secondary analysis that is initiated by and interacts with these contours to reconstruct the visible aspect of the surfaces (*Grossberg and Hong, 2006*; *Pessoa et al., 1995*). Although these models have proposed diffusion-like processes in retinotopic visual areas as a neural correlate for surface perception, feedback processes related to surface processing also display a delayed modulation of activity in early visual cortex of >100 ms (*Lamme, 1999*; *Self et al., 2013*). In addition, low-level aspects of the stimulus, such as the enhanced contrast at the edge and the absence of contrast inside the grey figure, can induce faster response latencies in early visual cortex at the edge representation compared to inside the homogeneous figure (*Albrecht et al., 2002*). Conceptually, an initial analysis of edges can also be seen as generating predictions for the presence of surfaces and their features, in line with the predictive coding hypothesis (*Rao and Ballard, 1999*). Hence, the earlier response to the edge compared to the surface is generally in line with a range of existing concepts and data about surface perception, but the question is whether and how this small temporal difference in neuronal responses translates into a ~ 2 s difference in BOLD response onset (see *Figure 5*). It is possible that the apparent delay in the onset of the BOLD response to the surface may be the result of competing positive and negative BOLD effects (*Uludağ and Blinder, 2018*). In the surface cortical representation, positive (due to luminance increase) and negative (due to lateral inhibition) BOLD responses may occur equally quickly and strongly, and hence may balance each other at the beginning of the stimulation. As time passes, the negative response may appear due to a more sustained negative response paired with a more transient or adaptive positive response. Thus, even though both the positive and negative BOLD responses may have similar latencies as the edge response, the sum of both centre responses may initially cancel out and lead to a larger apparent latency of the negative response emerging later on.

## Negative BOLD response

In line with a previous study (*Akin et al., 2014*), the surfaces yielded strongly negative BOLD responses in V1, V2 and V3, irrespective of whether they were perceived as static or moving (*Figure 3*). The negative response was located at the cortical retinotopic representation of the interior of the surface and was sustained throughout the presentation period (*Figure 5*, and *Figure 5—figure supplement 1*, *Figure 5—figure supplement 2*, *Figure 5—figure supplement 3*). A control experiment revealed that the negative response was only observed when the experimental stimuli were presented on a texture background (*Figures 6,7*). An additional control experiment showed that a full-screen texture pattern evokes a positive response when contrasted against a uniform rest condition (*Figure 7—figure supplement 1*). The absolute magnitude of the response to the full-screen texture was similar to the negative response observed in the main experiment (ca. 3% stimulus-induced signal change). In yet another control experiment, we investigated the effect of a texture background on stimulus-induced responses relative to a baseline with a uniform background

(*Figure 7—figure supplement 2*). The results lend further support to the hypothesis that the negative signal in the main experiment was driven by a relative lack of activation in response to a uniform stimulus compared to a texture surface. A simulation lends further support to this interpretation (*Figure 5—figure supplement 5*). Thus, we suggest that the negative surface response resulted primarily from an elevated baseline activity due to the texture background.

Note that the effect of the texture pattern was very strong. A change in the background from texture to homogeneous dark background resulted in a 4% signal change (from –3% to +1% BOLD). It is quite remarkable that a subtle change in the background leads to such a strong decrease in BOLD signal and presumably reduction in metabolism and excitatory neuronal activity. In comparison, Kok et al. observed a response amplitude of approximately 0.7% to 1.4% at the retinotopic representation of a centrally presented contrast-reversing checkerboard (using a similar MRI pulse sequence and the same spatial resolution as in the present study, *Kok et al., 2016*, see their Figure S2B; *Shmuel et al., 2002*; *Shmuel et al., 2006*).

Moreover, the negative BOLD response in the figure is not due to vascular steal: the current consensus is that vascular steal does not occur in healthy subjects, but is rather a sign of pathology, and is caused by a large, decrease in neural activation (*Boorman et al., 2010*; *Devor et al., 2007*; *Pasley et al., 2007*; e.g., see *Shmuel et al., 2002*, *Shmuel et al., 2006*). Instead, the results of our control experiments support the interpretation that the negative BOLD was due to the relative lack of stimulation in the grey figure region compared to the textured background.

## Relationship to electrophysiological findings

*Self et al., 2013* studied the laminar profile of figure-ground segregation in monkey V1. They observed neuronal activity related to feedforward, horizontal, and feedback mechanisms, that are thought to reflect processing of stimulus texture, borders, and figure-ground segregation, respectively. The feedback signals were strongest in superficial and deep layers (*Self et al., 2013*), in accordance with projection patterns observed in anatomical studies (*Anderson and Martin, 2009*; *Rockland and Pandya, 1979*; *Rockland and Virga, 1989*). The discrepancy with our results (top-down signal strongest towards superficial, but not deep, cortical depth in V1) could have several reasons. First, *Self et al., 2013* focus their analysis of the sustained parts of the response on multi-unit activity (MUA), which reflects local neuronal firing. In contrast, fMRI is most sensitive to postsynaptic activity (*Goense and Logothetis, 2008*; *Logothetis et al., 2001*; *Viswanathan and Freeman, 2007*). Second, some thalamo-cortical input from LGN targets V1 layer six rather than layer 4 (*Briggs and Usrey, 2007*; *Bullier and Henry, 1980*), and perhaps stimuli providing strong feedforward drive, such as the texture stimuli used by *Self et al., 2013* might have led to suprathreshold input to layer 6. There is some evidence for the presence of orientation sensitive signals in the LGN, and for a possible contributions of LGN to figure-ground segregation (*Kuhlmann and Vidyasagar, 2011*; *Poltoratski et al., 2016*; *Self and Roelfsema, 2015*; *Viswanathan et al., 2015*), which might help drive the differential figure-ground signal in deep V1 in *Self et al., 2013*. This thalamo-cortical interaction involving deep V1 possibly contributing to differential signals to figure and ground may be absent for the homogenous surface stimuli used in the present study, which can be expected to drive no or only weak feedforward drive (except for the edges). Third, *Self et al., 2013* used a stimulus that was behaviourally relevant – the monkeys were trained to perform a delayed saccade towards the stimulus. In contrast, the stimuli in our experiment were behaviourally irrelevant, and subjects were performing a central fixation task throughout the experiment. Furthermore, beyond the differences in signal measured, stimulus, and experimental design, it is difficult to use the few 100 ms typically measured post-stimulus onset in neurophysiological experiments as a predictor for fMRI activity measured 10 s and more after stimulus onset.

## Spatial deconvolution

A complicating factor in the analysis of the layered distribution of fMRI signal is related to the anatomy of ascending draining veins, which leads to a strong bias for the BOLD signal to be stronger in superficial cortical layers, even if the neuronal activity is stronger in deeper layers (*Koopmans et al., 2011*; *Markuerkiaga et al., 2016*; *Havlicek and Uludag, 2019*; see *Uludağ and Blinder, 2018* for a review). To use the BOLD signal as a realistic estimate of underlying neural activity in high-resolution data, it is therefore crucial to take this effect into account (*Markuerkiaga et al., 2016*). We have

previously employed a spatial deconvolution to remove signal spread due to ascending veins (*Marquardt et al., 2018*). The exact parameters of the spatial deconvolution are difficult to determine, and our parameter choices may not be exact. Nevertheless, simulations have shown that the spatial deconvolution is relatively robust against deviations in its model parameters (see *Marquardt et al., 2018*, *Figure 8*, and Supplementary Figures S4 & S5 therein). Although the exact shape of the resulting cortical depth profiles is contingent on the model parameters of the spatial deconvolution, the results do not differ qualitatively in case of different model parameters within physiologically plausible ranges (*Marquardt et al., 2018*). Thus, we stress the importance of data analysis, in general, and spatial deconvolution, in particular, for high-resolution fMRI to obtaining accurate representation of neuronal activity across cortical depths.

## Summary

Our study provides the first evidence that a motion percept in a surface region of a stimulus far removed from the local information inducing the motion percept produces a small increase in activity in the retinotopic representation of the figure. At the same time, our study reports a negative BOLD signal in the figure representation of an unexpected magnitude, and in contrast to standard expectation, following a luminance increase. This shows that subtle low-level aspects of the stimulus can have pronounced effects not only on the magnitude but even on the sign of the BOLD signal. It is an open question whether the neural mechanisms behind the negative response have a functional role in surface perception. In spite of the negative BOLD response, the perceptual assignment of a surface feature to a visual field region (where that feature was physically absent) yielded a signal enhancement, in line with other studies. While different surface features or displays may result in distinct depth resolved patterns of fMRI activity, possibly suggesting various sources of feedback, the consistent finding of signal enhancements during induced or illusory surface perception also suggests common aspects to the mechanisms of surface perception independent of the displays or features.

# Materials and methods

## Experimental design

Healthy participants (*n* = 9, age between 18 and 44 years, mean (SD) age 27.6 (7.3) years, four females, five males) gave informed consent before the experiment, and the study protocol was approved by the local ethics committee of the Faculty for Psychology and Neuroscience, Maastricht University (reference number: ERCPN 180_03_06_2017). Subjects were presented three visual stimuli: The main experimental stimulus was a 'Pac-Man' figure rotating around its centre (*Figure 1A*). There were two control conditions: First, the same Pac-Man figure as in the main condition was presented statically, that is. without rotating around its centre (*Figure 1B*). The second control stimulus consisted of a large, stationary wedge on the left side, and a smaller, rotating wedge on the right side (at the same location as the 'mouth' of the Pac-Man; *Figure 1C*). We will henceforth refer to these three conditions as 'motion induction stimulus', 'static control stimulus', and 'dynamic control stimulus', respectively. Note that in our figures, the texture backgrounds are proportionally reduced with stimulus size and do not convey a good impression of the granularity and contrast of the texture (e.g. compare *Figure 1—figure supplement 1* and *Figure 1—figure supplement 2*).

All three stimuli had a diameter of 7.5° visual angle. The 'mouth' of the Pac-Man had a circular arc of 70° (±35° from the right horizontal meridian). In the motion induction condition, the 'mouth' of the Pac-Man rotated clockwise and anticlockwise by ±35°, at a rate of 0.85 cycles per second. The angular position of the 'mouth' was modulated sinusoidally in order to create the impression of a smooth, natural, back and forth movement. In the dynamic control condition, the right-hand wedge rotated with the same frequency and angular displacement as the 'mouth' of the Pac-Man. The rotating, right-hand wedge had a circular arc of 65°, and the stationary, left-hand wedge had a circular arc of 220°. As a result, the motion induction stimulus is perceived to rotate clockwise and anticlockwise back and forth, whereas the dynamic control stimulus creates the impression of a rotating wedge on the right and a stationary wedge on the left. Importantly, the retinal image of all three stimuli is identical in the left visual field.

All stimuli were presented on a textured random noise background as was done in *Akin et al., 2014*, who included the texture to increase figure ground segregation. The background texture pattern was static, and was displayed throughout each run (i.e. also during rest periods). The texture pattern was created by randomly drawing pixel intensity values from a Gaussian distribution, and filtering the resulting image with a uniform kernel (kernel size 6 × 6 pixel). Before applying the uniform filter, the random Gaussian distribution of pixel intensities had a mean of 40 units and a standard deviation of 60 units (8-bit unsigned integer RGB pixel intensities, that is range 0 to 255). The granularity of the texture pattern is a function of the size of the filter kernel, and of the width of the Gaussian distribution, from which the pixel intensities are drawn. The relation between pixel intensity and luminance on our projection system was given by $y = -78.8 \times x^3 + 78.7 \times x^2 + 317.2 \times x + 163.3$, where $x$ represents the pixel intensity (in Psychopy convention, i.e. range –1.0 to 1.0), and $y$ corresponds to luminance (in cd/m$^2$). These values are based on measurements taken with a photometer (Konica Minolta CS-100A), and subsequent least-squares fitting of several functions, of which a third-degree polynomial provided by far the best fit. The mean luminance of the texture background was 8 cd/m$^2$, and the experimental stimuli (motion induction and control stimuli) had a uniform luminance of 163 cd/m$^2$. Videos of the stimuli are available online (https://doi.org/10.5281/zenodo.2583017).

Stimuli were created with Psychopy (*Peirce, 2007*; *Peirce, 2008*) and projected onto a translucent screen mounted behind the MRI head coil, via a mirror mounted at the end of the scanner bore. All lights in the scanner room were switched off during the experiment, and black cardboard was placed on the inside of the MRI transmit coil in order to minimise light reflection. The three stimulus conditions were presented in separate runs and in random order (see *Figure 1—figure supplement 2*). Stimuli were presented in a block design. In each run, there were 16 stimulus blocks with a fixed duration of 10.4 s, and 17 rest periods of variable duration in random order (possible durations were 18.7 s, 20.8 s, or 22.9 s). Each run began with an initial rest period with a fixed duration of 20.8 s, and ended with a rest period of one of the three possible durations. Each subject completed six functional runs (two for each stimulus condition; with the exception of one subject, who completed three repetitions each of the motion induction and dynamic control conditions, and two of the static control condition). The total duration of a run was 520 s, and there were 32 repetitions of each experimental stimulus in the main experiment (with the exception of one subject, for which there were 48 repetitions each of the motion induction and dynamic control conditions).

Participants were asked to fixate a central dot throughout the experiment and to report pseudo-randomly occurring changes in the dot's colour by button press. These targets were presented for 800 ms, with a mean inter-trial interval of 30 s (range ±10 s). No targets appeared during the first and last 15 s of each run. The timing of the colour changes was arranged such that the predicted haemodynamic responses to the experimental stimulus and to the colour changes are uncorrelated. To this end, a design vector representing the stimulus blocks and a design vector containing pseudo-randomly timed target events were separately convolved with a gamma function serving as model for the haemodynamic responses. The correlation between the predicted responses to the stimulus blocks and to the target events was calculated, and if the correlation coefficient was above threshold ($r > 0.001$), a new pseudo-random design matrix of target events was created. This procedure was repeated until the correlation was below threshold, separately for each run.

In an additional run, retinotopic mapping stimuli were presented for population receptive field estimation, allowing us to delineate early visual areas V1, V2, and V3 on the cortical surface (*Dumoulin and Wandell, 2008*). Please see section *Population receptive field mapping* (below) for details on the stimulus design of the population receptive field mapping paradigm.

In order to determine whether the responses are sustained or transient (*Horiguchi et al., 2009*; *Uludağ, 2008*), we acquired an additional experimental run for the motion induction condition with longer block durations in a subset of subjects ($n = 5$). The additional run had a duration of 424 s, during which the motion induction stimulus was presented five times for 25 s, interspersed between rest blocks of 50 s. As in the main experiment, subjects performed a central fixation task.

## Control experiment

A further control experiment was conducted to investigate the role of the stimulus shape and of the background in the processing of a surface stimulus. Two uniform surface stimuli were presented: A central disk from which a sector was removed (i.e. identical to the static control stimulus in the main experiment), and a central square. Both stimuli were identical in luminance and area. The square had

a side length of 6.65° visual angle. Both stimuli were presented under two background conditions: either on a uniform, dark grey background, or on a random texture background (same as in the main experiment). The two background conditions (i.e. uniform/texture) were presented in separate experimental runs, whereas the two stimulus shapes (i.e. Pac-Man/square) were presented in random order within runs. Stimulus blocks had a duration of 12.4 s, and were interspersed with variable rest blocks of 22.9 s, 25.0 s, or 27.0 s. The uniform background and the random texture pattern had a luminance of 8 cd/m$^2$, and the surface stimuli (Pac-Man and square) had a luminance of 163 cd/m$^2$ (same as in the main experiment). The control experiment was conducted in a separate session. Two subjects completed six experimental runs each (three with uniform background, three with texture background). Videos of the stimuli are available online (https://doi.org/10.5281/zenodo.2583017). As in the main experiment, retinotopic mapping runs were acquired in the same session.

An additional control experiment was performed to probe the response to a background texture in the absence of any additional stimulus (*Figure 7—figure supplement 1*). A uniform, dark grey background constituted the rest condition. The experimental stimulus was a full-screen, dark grey texture pattern (same as the background texture in the main experiment). The mean luminance was identical between the uniform rest condition and the texture stimulus. The texture stimulus was presented in blocks of ~12.5 s, interspersed with rest blocks of variable duration (~22.9 s,~25.0 s,~27.0 s). One subject completed four runs, with 12 repetitions of the full-screen texture stimulus.

## Data acquisition and preprocessing

Functional MRI data were acquired on a 7 T scanner (Siemens Medical Systems, Erlangen, Germany) and a 32-channel phased-array head coil (Nova Medical, Wilmington, MA, USA) using a 3D gradient echo (GE) EPI sequence (TR = 2.079 s, TE = 26 ms, nominal resolution 0.8 mm isotropic, 40 slices, coronal oblique slice orientation, phase encode direction right-to-left, phase partial Fourier 6/8; *Poser et al., 2010*). We also acquired whole-brain structural T1 images using the MP2RAGE sequence (*Marques et al., 2010*) with 0.7 mm isotropic voxels, and a pair of five SE EPI images with opposite phase encoding for distortion correction of the functional data (TR = 4.0 s, TE 41 = ms; position, orientation, and resolution same as for the GE sequence; *Feinberg et al., 2010*; *Moeller et al., 2010*; *Setsompop et al., 2012*).

Motion correction was performed using SPM 12 (*Friston et al., 1996*), and the data were distortion corrected using FSL TOPUP (*Andersson et al., 2003*). Since fMRI data were acquired using a 3D EPI sequence, no slice-time correction was applied. Standard statistical analyses were performed using FSL (*Smith et al., 2004*), fitting a general linear model (GLM) with separate predictors for the three stimulus conditions and a nuisance predictor for the target events of the fixation task. In order to account for both sustained and transient responses, each of the three stimulus conditions was modelled with two predictors: one based on a 'boxcar function' over the entire stimulus duration, and the other based on a delta function at stimulus onset and offset. (Only one predictor was used for the short target events.) All GLM predictors were convolved with a double-gamma haemodynamic response function. Highpass temporal filtering (cutoff = 35 s) was applied to the model and to the functional time series before GLM fitting. The parameter estimates obtained from the GLM were converted into percent signal change with respect to the initial pre-stimulus baseline (i.e. the first 20.8 s of each run). Throughout the manuscript, we use the term 'percent signal change' to refer to the relative signal change with respect to this pre-stimulus baseline. In case of differential contrasts between two stimulus conditions (*Figure 4*), we subtracted the percent signal change (relative to the respective pre-stimulus baseline) between the two conditions. Population receptive field mapping (*Dumoulin and Wandell, 2008*) was performed using publicly available python code (https://doi.org/10.5281/zenodo.1475439) and standard scientific python packages (Numpy, Scipy, Matplotlib, Cython; *Behnel et al., 2011*; *Hunter, 2007*; *Millman and Aivazis, 2011*; *Oliphant, 2007*; *van der Walt et al., 2011*). In order to facilitate reproducibility, the complete analysis pipeline was containerised within docker images (*Halchenko and Hanke, 2012*; *Kaczmarzyk et al., 2017*).

Cortical depth sampling requires a high level of spatial accuracy. In order to detect and remove low-quality data based on a quantifiable and reproducible exclusion criterion, we calculated the spatial correlation between each functional volume and the mean EPI image of that session after motion correction and distortion correction (see *Marquardt et al., 2018*, Supplementary Figure 1 therein, for details). If the mean correlation coefficient of the volumes in a run was below threshold ($r < 0.95$), that run would have been excluded from further analysis. However, no runs were excluded based on

the spatial correlation criterion. Moreover, it was important for subjects to be awake and to maintain fixation throughout the experiment. Therefore, runs in which subjects had detected less than 70% of targets were excluded from the analysis. This led to the exclusion of all runs from one subject. All other subjects had detected more than 70% of targets on all runs (mean hit rate for all subjects = 93%, standard deviation = 18%, mean hit rate after exclusion criterion = 98%, standard deviation = 5%).

## Segmentation and cortical depth sampling

Separately for each subject, the anatomical MP2RAGE images were registered to the mean functional image. In order to avoid downsampling of the anatomical images during registration, the mean functional image of each subject was upsampled to a resolution of 0.4 mm isotropic before registration (using trilinear interpolation). Thus, during registration of the anatomical images to the upsampled mean functional image, the anatomical images were indirectly upsampled (from 0.7 mm to 0.4 mm isotropic). This upsampling of anatomical images is beneficial for fine-grained tissue type segmentation, because it allows for better separation of adjacent sulci (avoiding erroneous grey matter 'bridges'). The anatomical images were roughly aligned in a first registration step based on normalized mutual information, followed by boundary-based registration (*Greve and Fischl, 2009*; *Jenkinson et al., 2002*; *Jenkinson and Smith, 2001*). The registered MP2RAGE images were used for tissue type segmentation. Initial tissue type segmentations was created with FSL FAST (*Zhang et al., 2001*). These initial segmentations were semi-automatically improved using the Segmentator software (*Gulban et al., 2018*) and ITK-SNAP (*Yushkevich et al., 2006*). These corrections of the segmentations obtained from FSL FAST were based on the T1 image from the MP2RAGE sequence, and aimed to remove mistakes in the definition of the white/grey matter boundary and at the pial surface.

The final white and grey matter definitions were used to construct cortical depth profiles using volume-preserving parcellation implemented in CBS-tools (*Bazin et al., 2007*; *Waehnert et al., 2014*). Specifically, the cortical grey matter was divided into 10 compartments, resulting in 11 depth-level images delineating the borders of these equi-volume compartments. The results from the GLM analysis, the population receptive field estimates, and event-related fMRI time courses were up-sampled to the resolution of the segmentations (i.e. 0.4 mm isotropic voxel size) using trilinear interpolation, and sampled along the previously established depth-levels using CBS-tools (*Bazin et al., 2007*; *Waehnert et al., 2014*). The depth-sampled data were projected onto a surface mesh (*Tosun et al., 2004*).

## ROI selection

We aimed to define ROIs in an observer-independent, quantifiable way. Only the first step of the ROI selection, that is the delineation of cortical areas V1, V2, and V3, was performed manually. The visual areas V1, V2, and V3 were delineated on the inflated cortical surface based on the polar angle estimates from the pRF modelling using Paraview (*Ahrens, 2005*; *Ayachit, 2015*). Subsequently, three selection criteria were applied for each location on the cortical surface for all cortical depths (i.e. each cortical segment) contained within V1, V2, or V3. First, only segments with good population receptive field model fits were included ($R2$ >0.15, median across cortical depth levels), excluding regions that are not specifically activated (e.g. possibly due to responses to a wide range of visual angles). Second, segments with low signal intensity in the mean EPI image were excluded, in order to avoid sampling from veins and low intensity regions around the transverse sinus, which may be present due to slight imprecisions in the registration and/or segmentation. Specifically, segments with a mean EPI image intensity below 7000 at any cortical depth (i.e. minimum over cortical depths) were excluded. (The mean EPI image intensity was ~10.000 for voxels within the brain.) Third, separate ROIs were defined for the centre of the stimulus, with eccentricities between 1° to 3° visual angle, and for the edge of the stimulus, at eccentricities between 3.5° and 4.0° visual angle (see *Figure 2*). The eccentricity of a segment was defined as the median eccentricity over cortical depths. The lower bound of the ROI corresponding to the stimulus centre was set to 1° (and not to 0°) in order to avoid the cortical representation of the fixation dot. Selection criteria were always applied to all cortical depths in a segment – that is either the entire cortical segment was included or excluded. Because the physically constant half of the stimulus was located in the left visual hemifield,

the analysis was restricted to the right hemisphere (with the exception of the visual field projections, which were reconstructed from both hemispheres; *Figures 3*,*7*). The ROI selection described in this section, and all subsequent analysis steps were performed using standard scientific python packages (Numpy, Scipy, Matplotlib; *Hunter, 2007*; *Millman and Aivazis, 2011*; *Oliphant, 2007*; *van der Walt et al., 2011*). Percent signal change values were averaged over the ROI, separately for each cortical depth level.

## Draining effect spatial deconvolution

Cortical depth-specific fMRI using GE sequences is affected by a venous bias caused by ascending draining veins, resulting in an fMRI signal increase towards the cortical surface (*Koopmans et al., 2011*; *Markuerkiaga et al., 2016*; see *Uludağ and Blinder, 2018* for a review; *Zhao et al., 2004*). In order to remove the effect of ascending veins from the cortical depth fMRI profiles, we employed leakage weights proposed by *Markuerkiaga et al., 2016*, and employed a spatial deconvolution approach described in detail in *Marquardt et al., 2018*. In brief, for each cortical depth level, we subtracted the estimated contribution of all deeper depth levels to obtain an estimate of the 'true' local signal change at that depth level.

## Visual field projection

While it is instructive to examine the spatial extent of activation on the inflated cortical surface, the exact relationship between the visual stimulus and the surface activation map is difficult to interpret: Cortical magnification and differences in receptive field size across the cortex complicate the mapping from visual space to the cortical surface. Therefore, we projected the activation maps into the visual field, based on population receptive field estimates. The resulting visual field projections reveal the spatial pattern of activation with respect to the stimulus-space. Population receptive field mapping (*Dumoulin and Wandell, 2008*) provides three parameters per vertex: x-position, y-position, and size of the Gaussian population receptive field model. For each vertex contained in the ROI, the 2D Gaussian population receptive field model was multiplied with the percent signal change for that vertex. The resulting scaled 2D Gaussians were summed over vertices. The result (a 2D array) was normalised by the population receptive field density at each visual field location (i.e. divided by the sum of 2D Gaussian over vertices).

More formally, let $M_{i,j,k}$ be a 3D tensor containing the population receptive field model for visual field positions $i, j$ for vertices $k$. The population receptive field model at each visual field location is a 2D Gaussian function:

$$\boldsymbol{M}_{i,j,k} = g(x_k, y_k, w_k)$$

where $x_k, y_k, w_k$ are the x-position, y-position, and width (standard deviation) of the 2D Gaussian for vertex $k$, respectively. Further, let $\boldsymbol{p}_k$ be a vector with percent signal change values for $n$ vertices contained in the ROI. The visual field projection ($V_{i,j}$) of percent signal change values ($\boldsymbol{p}_k$) was calculated as:

$$\boldsymbol{V}_{ij} = \frac{\sum_{k=1}^{n} \boldsymbol{M}_{i,j,k} \odot \boldsymbol{p}_k}{\sum_{k=1}^{n} \boldsymbol{M}_{i,j,k}}$$

where the multiplication and division operations are element-wise. The visual field projection $\boldsymbol{V}_{i,j}$ was calculated separately for each ROI and cortical depth level, but together for all subjects (by concatenating all subjects' population receptive field models, $\boldsymbol{M}_{i,j,k}$, and percent signal change vectors, $\boldsymbol{p}_k$). In this way, all subjects' activation maps can be projected into a single visual space; this is essentially a simple form of 'hyperalignment'. (The procedure is similar to that employed by *Kok et al., 2016*, with the difference that we did not apply any smoothing to the visual field projection.)

## Hypothesis testing

Differences in stimulus-induced activation were investigated by means of a linear mixed effects model. First, we assessed whether the stimuli differentially activated brain areas V1, V2, and V3. (In

other words, did activation differ between ROIs as a function of condition?) Second, we tested whether the activation profiles across cortical depth differed between brain areas. Both tests were implemented by means of a mixed effects model including the fixed factors ROI, stimulus condition, and cortical depth, and a random slope for subjects. The autocorrelation structure of cortical depth (within subjects) was modelled as continuous autoregressive of order one. For the first test, a model with all possible two-way interactions was compared with a null model, from which the stimulus condition by ROI interaction had been omitted (because this interaction reflects a differential effect of stimulus condition on brain areas). The second test compared a model with all possible two-way interactions with a null model without the cortical depth by ROI interaction (reflecting differences in cortical depth profiles between areas). The mixed effects models were fitted based on the percent signal change estimate of the sustained and transient predictors (for the stimulus centre and edge, respectively) obtained from the GLM. Comparisons of the respective pairs of models were conducted with a likelihood ratio tests. Models were fitted and compared using R and the nlme package (*Pinheiro et al., 2017*; *R Development Core Team, 2017*).

We investigated the shape of cortical depth profiles in more detail by comparing the distribution of peak positions in superficial layers between cortical areas. The position of the peak in the cortical depth profile of a condition contrast (i.e. experimental vs. control condition) indicates at which cortical depth the effect of the experimental manipulation was strongest. (We defined the peak position as the global maximum of the cortical depth profiles, because in our data there were no local maxima apart from the global maxima; *Figure 4—figure supplement 3*). A peak was counted as 'superficial' if it was located in the upper third of the grey matter (i.e. within 33% cortical depth relative to the CSF border). The ratio of superficial peaks (in the single subject profiles) was compared between areas with a chi-squared test.

## Population receptive field mapping

Stimuli used for population receptive field mapping were oriented bars at four different orientations and eight different positions per orientation, containing a black and white checkerboard pattern. The bars had a width of 1.25° visual angle, and the carrier pattern within the bar had a spatial frequency of 1.2 cycles/deg. The luminance of the black and white sectors of the carrier pattern was 2 $cd/m^2$ and 1390 $cd/m^2$, respectively, resulting in a luminance contrast of ~1. The polarity of the checkerboard pattern was reversed at a frequency of 4 Hz, and the bar changed its position every 2.079 s (in synchrony with the volume TR). Each of the resulting 32 stimulus configurations was presented 12 times for 2.08 s in random order. The duration of the population receptive field mapping run was 832 s (400 volumes). Similar to the main experiment, subjects were instructed to perform a central fixation task during the retinotopic mapping experiment. The software used for the presentation of the retinotopic mapping stimuli is publicly available (https://doi.org/10.5281/zenodo.1475439).

## Acknowledgements

This work was financially supported by funding from IBS (\#IBS-R015-D1) to KU and the Netherlands Organization for Scientific Research (NWO; Research Talent 406-14-085) to KU and IM.

## Additional information

### Funding

| Funder | Grant reference number | Author |
| --- | --- | --- |
| Nederlandse Organisatie voor Wetenschappelijk Onderzoek | 452-11-002 | Kâmil Uludağ |
| Nederlandse Organisatie voor Wetenschappelijk Onderzoek | 406-14-085 | Ingo Marquardt Kâmil Uludağ |
| Institute for Basic Science | IBS-R015-D1 | Kâmil Uludağ |

The funders had no role in study design, data collection and interpretation, or the decision to submit the work for publication.

## Author contributions
Ingo Marquardt, Conceptualization, Data curation, Software, Formal analysis, Funding acquisition, Investigation, Visualization, Methodology, Writing - original draft, Writing - review and editing; Peter De Weerd, Conceptualization, Formal analysis, Supervision, Methodology, Writing - original draft, Writing - review and editing; Marian Schneider, Conceptualization, Software, Investigation, Methodology; Omer Faruk Gulban, Conceptualization, Software, Methodology; Dimo Ivanov, Investigation; Yawen Wang, Formal analysis, Investigation, Visualization; Kâmil Uludağ, Conceptualization, Supervision, Funding acquisition, Methodology, Writing - original draft, Project administration, Writing - review and editing

## Author ORCIDs
Ingo Marquardt (iD) https://orcid.org/0000-0001-5178-9951
Peter De Weerd (iD) https://orcid.org/0000-0003-2252-5548
Marian Schneider (iD) https://orcid.org/0000-0003-3192-5316
Omer Faruk Gulban (iD) https://orcid.org/0000-0001-7761-3727
Yawen Wang (iD) https://orcid.org/0000-0003-0002-0768
Kâmil Uludağ (iD) https://orcid.org/0000-0002-2813-5930

## Ethics
Human subjects: Healthy participants gave informed consent before the experiment, and the study protocol was approved by the local ethics committee of the Faculty for Psychology & Neuroscience, Maastricht University. (reference number: ERCPN 180_03_06_2017 ).

## Decision letter and Author response
Decision letter https://doi.org/10.7554/eLife.50933.sa1
Author response https://doi.org/10.7554/eLife.50933.sa2

# Additional files

## Supplementary files
• Transparent reporting form

## Data availability
The fMRI dataset, experimental stimuli, and analysis code are publicly available. The fMRI dataset is available on Zenodo (https://doi.org/10.5281/zenodo.3366301). The software used for the presentation of retinotopic mapping stimuli, and for the corresponding analysis, is available on github (https://github.com/ingo-m/pyprf). Example videos of the main experimental stimuli are available on Zenodo (https://doi.org/10.5281/zenodo.2583017). If you would like to reproduce the experimental stimuli, the respective PsychoPy code can be found on github (https://github.com/ingo-m/PacMan/tree/master/stimuli/experiment). The respective repository also contains the analysis code and a brief description how to reproduce the analysis (https://github.com/ingo-m/PacMan; copy archived at https://github.com/elifesciences-publications/PacMan). High-level visualisations (e.g. cortical depth profiles & signal timecourses) and group-level statistical tests are implemented in a separate repository (https://github.com/ingo-m/py_depthsampling/tree/PacMan; copy archived at https://github.com/elifesciences-publications/py_depthsampling).

The following dataset was generated:

| Author(s) | Year | Dataset title | Dataset URL | Database and Identifier |
|---|---|---|---|---|
| Marquardt I, De Weerd P, Schneider M, Gulban OF, Ivanov D, Uludağ K | 2019 | Dataset: Feedback contribution to surface motion perception in the human early visual cortex | https://doi.org/10.5281/zenodo.3366301 | Zenodo, 10.5281/zenodo.3366301 |

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
