## [Decision Letter]

**Acceptance summary:**

This paper showcases the use of ultra-high field (7 Tesla) fMRI for disentangling the feedforward and feedback interactions across the cortical hierarchy underlying perception. The authors perform a detailed quantification of the laminar-specific patters of neural activity in human early visual cortex (areas V1-V3) during motion-based surface segmentation. This allows them to identify a signal component associated with illusory motion perception that is consistent with feedback from a higher-tier cortical area.

**Decision letter after peer review:**

Thank you for submitting your article "Feedback contribution to surface motion perception in the human early visual cortex" for consideration by *eLife*. Your article has been reviewed by three peer reviewers, and the evaluation has been overseen by Tobias Donner as the Reviewing Editor and Christian Büchel as the Senior Editor.

The following individuals involved in review of your submission have agreed to reveal their identity: Tomas Knapen (Reviewer #1); Lars Muckli (Reviewer #3).

The reviewers have discussed the reviews with one another and the Reviewing Editor has drafted this decision to help you prepare a revised submission.

Summary:

This study investigates the role of feedback in surface motion perception. The authors use high-field (7T) fMRI to distinguish which layers of the early visual cortex are activated by global motion perception of a surface. This study provides new insights into neuronal processes underlying perception of surface motion – an important and ubiquitous perceptual phenomenon. The perception of surfaces is a construction taking into consideration nearby contextual information, in this instance, cues that give rise to perception of surface motion. The main novel contribution lies in the fact that the authors separate center from surface responses, which has not been done in humans before. Reviewers applaud the authors for approaching important aspects of high-resolution fMRI of visual cortex (anatomical segmentation, unwarping, registration, pRF-based retinotopy) with great care and state-of-the-art methodology. However, reviewers also raised a number of substantive concerns, which would need to be addressed in revision.

Essential revisions:

1) Address the negative overall BOLD response.

The responses in the surface region are dominated by a large negative BOLD response. This negative BOLD effect seems to dwarf the cortical depth-dependent results by a factor of around 20. Reviewers wondered whether a less negative BOLD response for the dynamic Pacman condition could be the result of 1) an increase in activity, or 2) a decrease in negative evoked activity.

In general, the physiological basis of negative BOLD is not well understood. Is it possible that activation in the right visual field could cause negative BOLD in the right V1, not because it induces neural activity in right V1 but for haemodynamic reasons (e.g. “blood stealing”) that are not well understood? In this case, given that right visual field stimulation is different in the three different conditions, could there be different amplitudes of negative BOLD in right V1 due to this? In other words: can we assume that any BOLD effect in right V1 is not influenced by neural activity in left V1? This could present a problem in the interpretation of the current results, which would need to be addressed. Also, reviewers thought that the presence of the big negative BOLD response creates a discrepancy in the paper, which focusses on the depth-dependent results.

Reviewers felt the study would benefit substantially from testing whether the depth-dependent results hold even when the sign of the figure response is changed. This could be achieved by manipulating the background and inverting the sign of the figure-related BOLD response – the feedback-related findings should be the same in the different background conditions.

After some discussion, reviewers and Reviewing Editor agreed that you could approach this general issue in one of two ways:

i) You run an additional experiment that repeats the figure segmentation experiment without the textured background, so as to avoid the large negative BOLD response.

ii) You discuss this issue more prominently in the paper, and acknowledge the resulting limitations in terms of interpretation in the Discussion.

While option (i) would be the preferred course of action (allowing for a conclusive answer to this point), we (and reviewers) leave it to you to decide which of those options you choose.

2) Present laminar results in more detail; in particular, show individual subject data.

The depth-dependent results are presented relatively minimalistically and the analysis is done in summary form using linear mixed models. Two specific suggestions for improvement:

a) It seems that he peak of the depth-level BOLD response is shifted towards CSF for V1 relative to V2 and V3. Can the authors also make this point by an analysis that stays a little closer to the data, for example (and this is just an example) by fitting a quadratic function to the depth-level bold responses and showing that the peak indeed shifts upward for V1?

b) Layer-specific fMRI effects are still a challenging field. To improve the interpretation it would be good to show individual results – alignment of the depth layers to the activity – and main effects of profile across subjects (as supplementary figure).

3) Discuss relationship to (discrepancies with) animal physiology.

Similar questions have been addressed with electrophysiological recordings in animals, particularly by Pieter Roelfsema's lab. In the current study, you report that the laminar profile of surface segmentation differs across the different regions of the early visual cortex, with the effect peaking in superficial layers in V1, and in middle layers in V2 and V3. You propose a neural mechanism for these profiles, with top-down feedback targeting the superficial layers of V1, which subsequently target the middle layers of V2 and (directly or indirectly) V3.

These findings differ from effects of surface segmentation using electrophysiology in animals, which report both superficial and deep layers of V1 being involved. Please discuss possible sources of this apparent discrepancy.

4) Address the role of attention.

A similar interpretational problem as with the negative BOLD issue (point 1) arises with the role of attention. Attention has been suggested to have a similar laminar profile (Lawrence et al., 2019). Also note that Akin et al. suggest that the effect reported here is dependent on attention. It seems possible that the current signals do not at all reflect the neural correlate of surface segmentation, but rather only the secondary effect of top-down attention directed at a (perceptually moving) segmented surface? If the reported signal is an exclusively attentional one, rather than the perceptual signal that captured attention, this would change the interpretation quite profoundly.

As with point 1, the most direct way of addressing of this issue would be to run an additional experiment that controls attention more effectively than the current fixation task – either using a more challenging fixation task and showing psychophysically that this puts the stimuli of interest in the "near-absence" of attention (see work by Jochen Braun, Christof Koch, David Heeger, Sang-Hun Lee, and others); or manipulate attentional load parametrically and show that the current signals are (largely) independent of this. the alternative would be to acknowledge this issue and the resulting limitations in terms of interpretability. Again, we leave the decision about the course of action to you; but please note that points 1) and 2) could be addressed in a single experiment.

[Editors' note: further revisions were suggested prior to acceptance, as described below.]

Thank you for submitting your article "Feedback contribution to surface motion perception in the human early visual cortex" for consideration by *eLife*. Your article has been reviewed by two peer reviewers, and the evaluation has been overseen by Tobias Donner as the Reviewing Editor and Christian Büchel as the Senior Editor. The following reviewer has agreed to reveal their identity: Tomas Knapen (Reviewer #1).

The reviewers have discussed the reviews with one another and the Reviewing Editor has drafted this decision to help you prepare a revised submission.

We would like to draw your attention to changes in our revision policy that we have made in response to COVID-19 (https://elifesciences.org/articles/57162). Specifically, when editors judge that a submitted work as a whole belongs in *eLife* but that some conclusions require a modest amount of additional new data, as they do with your paper, we are asking that the manuscript be revised to limit claims to those supported by data in hand, or to explicitly state that the relevant conclusions require additional supporting data.

Summary:

This is an overall well-conducted high-resolution fMRI study on the cortical mechanisms of visual surface perception. The authors have thoroughly revised their work based on the reviewer comments. Reviewers and Editors feel that most of the comments were addressed adequately, with some additional control data and additional discussion of the interpretation of the effects. This has improved the manuscript. Yet, there do remain some lingering issues, which we feel would need to be addressed more explicitly through textual revision before this paper can be accepted for publication at *eLife*. We feel the authors should invest more into incorporating the reviewer comments, rather than simply defending their original conclusions.

1) Relationship with attention.

The authors have opted not to run an additional control experiment, but rather address the potential role of attention in their findings in Discussion. That is fine in principle, but we do find the new paragraph to be unsatisfying and unclear. Perhaps there may be some confusion about the nature of the concern. We realize that attention may play an intricate (perhaps necessary) role in the cortical computation underlying surface segmentation. But that is not the issue here: the issue is whether the reported effects are (i) due to this computation per se (whether or not some form of attention is involved), or (ii) due to a purely *secondary* effect of attention being drawn to a segmented surface.

This matters for how the results are conceptualized and understood by the broad readership: The first would constitute a specific neural correlate of surface segmentation per se, whereas the second would be a non-specific signature of any form of object perception (compared to no object).

We acknowledge that arbitrating between these two scenarios is generally a hard problem in visual neuroscience; yet, there are a few compelling demonstrations in the literature for doing so. We feel that, without using similar manipulations here, there is just now way of knowing, which of these two scenarios accounts for the current results. Future experiments will be needed to pinpoint this.

These points, and in particular the two possible scenarios, should be made explicit in the new Discussion paragraph.

2) Negative BOLD response due to stimulus with textured background.

We remain puzzled by the negative BOLD. Intriguingly, the onset of a whole-field textured background (Figure 6—figure supplement 3) elicits an fMRI response that is comparable in positive amplitude to the negative effects within the stimulus region (Figure 4 and 7). This might indicate that in the main experiment, the textured background causes ongoing positive responses throughout the run, presumably maintained by ongoing micro saccades and fixational instability. Then, the presentation onset of a stimulus has a two-fold effect; it decreases the ongoing response to this background (because it disappears) while also playing the role of an activating stimulus very locally (dark patches in the texture are brighter, whereas light patches are darker, stimulating both on and off visual channels temporarily). In this scenario, the negative response shown in Figure 4 should be seen as a combination of a positive and a negative component, that summed together produce the appearance of a delayed negative response.

If the negative BOLD response is indeed a composite of a positive and negative BOLD response, it would be interesting to see how the laminar effects can perhaps decompose this composite. A laminar analysis conducted on the first and later parts of the response separately may be highly insightful here. We understand that SNR may be insufficient to conclusively accommodate this analysis.

This idea (possibly others) deserves unpacking, over and above the brief hint towards this possibility the authors mention in subsection “Background dependence of the negative response”. The fact that the authors have seen similar responses in other contexts doesn't necessarily speak to mechanism in this case.

3) Relationship/discrepancy with previous animal neurophysiology studies:

The authors state in rebuttal: "Finally, electrophysiological experiments typically measure responses over a time window of only a few hundred milliseconds after stimulus onset (300 ms in case of Self et al., 2013). However, the initial and the sustained response to a surface stimulus can have different laminar profiles in V1 (Maier et al., 2011)."

If anything, this makes the discrepancy more puzzling: sustained responses, which the authors imply are more strongly reflected in the BOLD response, are predominant in the deep, not the middle and superficial layers (Maier et al., 2011; Self et al., 2013). This part of the Discussion needs revision.

4) Possibility of visual cortex effects being the result of top-down feedback to the LGN being fed forward to V1.

The authors seem to simply dismiss this possibility, which we find puzzling; a previous study (Akin et al) showed that the experimental modulation used here affects LGN, and depth dependent profiles as reported here in Figure 3, i.e. mostly middle and superficial activations, seems to be very plausibly in line with the effect arising from the LGN, especially when taking the deep-to-superficial draining vein effects into account. Of course, the authors performed a deconvolution to address drain, but we cannot assume that this got rid of any draining vein effects completely. Again, this alternative explanation is worth discussing, rather than simply dismissing it.

---

## [Author Response]

Essential revisions:1) Address the negative overall BOLD response.The responses in the surface region are dominated by a large negative BOLD response. This negative BOLD effect seems to dwarf the cortical depth-dependent results by a factor of around 20. Reviewers wondered whether a less negative BOLD response for the dynamic Pacman condition could be the result of 1) an increase in activity, or 2) a decrease in negative evoked activity.In general, the physiological basis of negative BOLD is not well understood. Is it possible that activation in the right visual field could cause negative BOLD in the right V1, not because it induces neural activity in right V1 but for haemodynamic reasons (e.g. “blood stealing”) that are not well understood? In this case, given that right visual field stimulation is different in the three different conditions, could there be different amplitudes of negative BOLD in right V1 due to this? In other words: can we assume that any BOLD effect in right V1 is not influenced by neural activity in left V1? This could present a problem in the interpretation of the current results, which would need to be addressed. Also, reviewers thought that the presence of the big negative BOLD response creates a discrepancy in the paper, which focusses on the depth-dependent results.Reviewers felt the study would benefit substantially from testing whether the depth-dependent results hold even when the sign of the figure response is changed. This could be achieved by manipulating the background and inverting the sign of the figure-related BOLD response – the feedback-related findings should be the same in the different background conditions.After some discussion, reviewers and Reviewing Editor agreed that you could approach this general issue in one of two ways:i) You run an additional experiment that repeats the figure segmentation experiment without the textured background, so as to avoid the large negative BOLD response.ii) You discuss this issue more prominently in the paper, and acknowledge the resulting limitations in terms of interpretation in the Discussion.While option (i) would would be the preferred course of action (allowing for a conclusive answer to this point), we (and reviewers) leave it to you to decide which of those options you choose.

We thank the reviewers for their thorough assessment of our manuscript. We agree that follow-up experiments investigating the negative response (and also the temporal dynamics of the response, especially at the stimulus edge) would be very interesting, but we think that the current data by itself provides important novel insights to justify publication, and discuss the insights and limitations of this study in detail. Moreover, we present new data from limited additional control experiments.

Your remark comprises several important considerations that we would like to address in turn:

1) The relatively low effect size of the top-down motion effect, relative to the amplitude of the negative BOLD response

2) The physiological basis of the negative BOLD response

3) A possible interhemispheric interaction

1) The small effect of the top down effect

The amplitude of the negative response (caused by the texture background) is indeed large, compared to the smaller top-down, apparent-motion effect. However, such a discrepancy in the magnitude of (presumably) bottom-up and top-down effects is not uncommon. We have added the following paragraph to the Discussion section to address this issue. (See also following point, related to the interpretation of negative response.)

“The amplitude of the apparent motion effect (Figure 3) is small, compared with the overall response strength (e.g. Figure 4). […] We see no principled objection against the interpretability of a condition contrast among control and experimental responses that both yield BOLD responses smaller than the BOLD baseline response.”

2) Negative BOLD

To better understand the negative response to the figure in our experiments, we conducted a few limited post-hoc experiments in single participants. First, we contrasted the texture against an equiluminant, dark background in a single participant. This experiment showed that the texture yields a strong, positive response. The methods for this experiment are described in “Control experiment”, results presented in paragraph one of the Results and subsection “ Background dependence of the negative response”, and discussed in sub-section “Negative BOLD response”.

It should also be mentioned that the reduced size of the stimuli in the figures, gives a somewhat misleading idea on the visibility of the texture. Figure 1—figure supplement 1, if rendered such that the radius of the disk is 3.75cm gives, an impression of what it would have looked like in the scanner, when viewed from ~57cm. The clear granularity of the texture with peak-to-peak Michelson contrast of 92% is in line with a significant texture-driven response, which could be the basis of the negative BOLD responses in our experiments.

At the end of first paragraph of Materials and methods section, we have added a sentence to make sure the readers have a correct impression of the stimulus and its background: “Note that in our figures, the texture backgrounds are proportionally reduced with stimulus size and do not convey a good impression of the granularity and contrast of the texture (e.g. compare Figure 1—figure supplement 1 and Figure 1—figure supplement 2).”

In a further experiment (see new Figure 6—figure supplement 3), we presented one participant with the following conditions: Baseline condition (homogenous dark); Condition A grey square on black homogenous background; Condition B grey square on black textured background; Condition C whole field texture background. The fixation spot was placed at the centre of the screen, the square was 6.95 degree x 6.95 degree in size, with its centre at a 3.82 deg eccentricity. Analysis was done in a ROI corresponding to the centre of the square representation in V1.

As in the previous control experiment, contrasting full-field texture (C) with equiluminous homogenous black (Baseline) gave a strong positive BOLD response in the square ROI, similar to the negative BOLD in the conditions of our main experiment. There was also a limited positive BOLD response when contrasting the grey square on texture (B) to the grey square on homogenous black (A), suggesting some leakage of the texture-driven response into the square representation. In addition, subtracting activity for the square on texture (B) from full-field texture (C) yielded a negative BOLD response, again similar to what was observed in our main experiment. Taken together, these results suggest that for the stimuli we have chosen, a straightforward explanation of the negative BOLD inside the figure is the relative lack of activation driven by a homogenous stimulus compared to a texture surface.

The BOLD response in the figure hence is not due to vascular steal: the current consensus is that vascular steal does not occur in healthy subjects and is rather a sign of pathology, and is caused by a large, decrease in neural activation (e.g. see Shmuel et al., 2002; Pasley et al., 2007; Shmuel et al., 2006; Boorman et al., 2010; Devor et al., 2007). Instead the results support the interpretation that the negative BOLD was due to the relative lack of stimulation in the grey figure region compared to the textured background.

The following sentence was added to the first paragraph of the Results section, to make clear to the reader early on that the texture can explain the negative response in the main experiment:

“Control experiments supported the idea that the negative sign of the response was related to the much stronger response to the texture in the background than to the homogenous grey in the figure (see Figure 6—figure supplement 2 and Figure 6—figure supplement 3).”

Moreover, the following paragraph was added to the Discussion subsection “Negative BOLD response”:

“A control experiment revealed that the negative response was only observed when the experimental stimuli were presented on a texture background (Figure 6 and 7). […] Instead, the results of our control experiments support the interpretation that the negative BOLD was due to the relative lack of stimulation in the grey figure region compared to the textured background.”

3) Differences in right-field stimulation affecting interhemispheric balance and contributing to the correlate of the illusion

Our design used a stimulus manipulation in the right hemifield, in order to induce an illusion in the left hemifield. This raises the possibility of interhemispheric interactions that may cause a stronger negative BOLD signal in the illusory condition compared to the control condition. In principle, a combination of transcallosal, interhemispheric connections (Clarke and Miklossy, 1990; Essen and Zeki, 1978; Glickstein and Whitteridge, 1976; Houzel and Milleret, 1999; Van Essen et al., 1982; Wong-Riley, 1974) as well as feedback of lateral interactions taking place in higher-order visual cortex could affect the interhemispheric balance of activity in V1, V2 and V3. These higher level interactions may also include attentional imbalances. Several studies have demonstrated that spatial attention causes both a positive response in the cortical representation of attended locations, and a negative response at unattended locations (Bressler et al., 2013; Müller and Kleinschmidt, 2004; Silver et al., 2007; Slotnick et al., 2003; Somers et al., 1999; Tootell et al., 1998).

As we indicated above, we propose that the primary cause of the negative BOLD in the grey surface is primarily related to the absence of texture inside the figure compared to the texture baseline. Nevertheless, we verified whether an imbalance between the constant area of the stimulus in one hemifield (where the illusion is probed), and the part of the stimulus in the other hemifield (where the illusion is either generated or prevented in the control conditions) could explain the pattern of results we obtained.

Our data directly speak against this possibility. The right field stimulation in the motion illusion stimulus (Figure 1—figure supplement 2A) and the dynamic control stimulus (Figure 1—figure supplement 2C) were designed to be similarly high, whereas the right field stimulation in the static control condition (Figure 1—figure supplement 2B) was designed to be low. If an activity imbalance between hemifields had been the primary cause of our pattern of results, we would have expected that the dynamic control condition and the illusion condition would both have induced a similar decrease in activity in the left hemifield representation compared to the static control condition. In addition, in the left hemifield’s representation, an increased (less negative) BOLD would have been observed for the dynamic than for the static control condition. Instead, we find that in V1 and V2, the static and dynamic control conditions yield the same activity levels in the left hemifield representation, and that in the left hemifield representation the motion illusion condition gives the *highest* (least negative) response, i.e., the illusory motion condition yields activity that exceeds (i.e. is less negative than) the activity in both control conditions (see Figure 3—figure supplement 1). This shows that the dynamic stimulation in the right hemifield in itself (and a potential associated imbalance in attention) is unlikely to be responsible for the observed pattern of results in V1 and V2.

Note that in V3, we found that the response in the representation of the left part of the stimulus (where the illusion is probed) is smaller for the dynamic control condition than for the static control condition, which suggests that V3, in contrast to V1 and V2, is sensitive to differences in activity between hemifields. Accordingly, when contrasting the motion illusion condition against the dynamic control condition, the positive difference is smaller than when contrasting the motion illusion condition against the static control condition. This shows that, in V3, the former rather than the latter statistical contrast provides the best estimation of the motion illusion, and that overall the dynamic control condition was superior over the static control condition.

We have included the above four paragraphs integrally in a new sub-section in the Discussion (“Hemispheric imbalances in stimulation”). Also note that we have backed up the last paragraph, which is based on a difference in activity levels in the left-hemisphere part of the stimulus between static and dynamic control conditions, by additional analysis that is included in the Results section. We slightly changed the flow on these two pages to be able to include the results of these extra analyses.

Related to the apparent difference between the control conditions in V2 and V3, we ran a mixed-effects model comparison, but restricted to V2 and V3 and the two control conditions. We found a significant ROI by condition interaction (likelihood ratio (df): 16.7 (1), p < 0.0001). In other words, we found a statistically significant difference in the pattern of stimulus-induced activation caused by the control conditions in V2 and V3. Moreover, we found a ROI by depth interaction showing that the depth profiles differed between ROIs (likelihood ratio (df): 3.9 (1), p = 0.0491).

Another relevant outcome of the anatomically restricted analysis was that in V3, the dynamic and stationary controls were not equivalent (mixed effects model comparison, limited to V3 and the two control conditions, testing for an effect of “condition”; likelihood ratio (df): 35.8 (1), p = <0.0001). We included this result as well as it further supports the fact that the depth resolved activity profiles were different among visual areas.

2) Present laminar results in more detail; in particular, show individual subject data.The depth-dependent results are presented relatively minimalistically and the analysis is done in summary form using linear mixed models. Two specific suggestions for improvement:a) It seems that he peak of the depth-level BOLD response is shifted towards CSF for V1 relative to V2 and V3. Can the authors also make this point by an analysis that stays a little closer to the data, for example (and this is just an example) by fitting a quadratic function to the depth-level bold responses and showing that the peak indeed shifts upward for V1?

Thank you for pointing out this issue, we agree that a more direct test of the distribution of peak positions is required. First, we include a new Figure with single subject cortical depth profiles (Figure 3—figure supplement 2). Second, we provide an additional statistical test. As you point out, the question we would like to address is whether the distribution of peak positions is shifted towards superficial layers in V1, relative to V2 and V3. We propose to address this question by a chi-square test on the distribution of superficial peaks, as follows (see new paragraphs in Materials and methods and Results sections respectively):

Materials and methods:

“We investigated the shape of cortical depth profiles in more detail by comparing the distribution of peak positions in superficial layers between cortical areas. […] The ratio of superficial peaks (in the single subject profiles) was compared between areas with a chi-squared test.”

Results

“The ratio of superficial peak positions in the cortical depth profiles of the condition contrast “motion induction stimulus” vs. “dynamic control stimulus” was compared with a chi-squared test. […] Thus, the ratio of superficial peak positions (in the single subject cortical depth profiles) is significantly higher in striate than in extrastriate cortex.”

b) Layer-specific fMRI effects are still a challenging field. To improve the interpretation it would be good to show individual results – alignment of the depth layers to the activity – and main effects of profile across subjects (as supplementary figure).

We agree that it is good practice to be as transparent as possible, and particularly with a relatively novel technique as layer-specific fMRI. The updated Figure 3—figure supplement 2 presents single-subject cortical depth profiles. The single-subject profiles are noisy, in line with the relatively low SNR of the depth-specific fMRI signal, as can also be seen in previous publications (see e.g. Muckli et al., 2015; Kok et al., 2016. Figure S3).

3) Discuss relationship to (discrepancies with) animal physiology.Similar questions have been addressed with electrophysiological recordings in animals, particularly by Pieter Roelfsema's lab. In the current study, you report that the laminar profile of surface segmentation differs across the different regions of the early visual cortex, with the effect peaking in superficial layers in V1, and in middle layers in V2 and V3. You propose a neural mechanism for these profiles, with top-down feedback targeting the superficial layers of V1, which subsequently target the middle layers of V2 and (directly or indirectly) V3.These findings differ from effects of surface segmentation using electrophysiology in animals, which report both superficial and deep layers of V1 being involved. Please discuss possible sources of this apparent discrepancy.

We agree that this aspect deserves more attention, and have expanded the Discussion section accordingly, as follows:

“Relationship to electrophysiological findings

Self et al. (Self et al., 2013) studied the laminar profile of figure-ground segregation in monkey V1. They observed neuronal activity related to feedforward, horizontal, and feedback mechanisms, that are thought to reflect processing of stimulus texture, borders, and figure-ground segregation, respectively. […] Results based on experimental designs similar to the one used in the present study have, to the best of our knowledge, not been reported.”

4) Address the role of attention.A similar interpretational problem as with the negative BOLD issue (point 1) arises with the role of attention. Attention has been suggested to have a similar laminar profile (Lawrence et al., 2019). Also note that Akin et al. suggest that the effect reported here is dependent on attention. It seems possible that the current signals do not at all reflect the neural correlate of surface segmentation, but rather only the secondary effect of top-down attention directed at a (perceptually moving) segmented surface? If the reported signal is an exclusively attentional one, rather than the perceptual signal that captured attention, this would change the interpretation quite profoundly.As with point 1, the most direct way of addressing of this issue would be to run an additional experiment that controls attention more effectively than the current fixation task – either using a more challenging fixation task and showing psychophysically that this puts the stimuli of interest in the "near-absence" of attention (see work by Jochen Braun, Christof Koch, David Heeger, Sang-Hun Lee, and others); or manipulate attentional load parametrically and show that the current signals are (largely) independent of this. the alternative would be to acknowledge this issue and the resulting limitations in terms of interpretability. Again, we leave the decision about the course of action to you; but please note that points 1) and 2) could be addressed in a single experiment.

The overall magnitude of the negative BOLD is more likely explained by the absence of texture in the figure region. The small differences in activity (less negative BOLD for induced motion compared to control conditions) reflect the differences among the experimental and control stimuli. With respect to attention, we have excluded one major factor related to potential imbalances in stimulation and attention to the left and right halves of the stimulus display (see point 1). In addition, a comparison with results in a recent paper by Lawrence et al., 2018, suggests that the depth distribution of activity in our data did not reflect a mere attentional effect. In Lawrence et al., 2018, attention to a stationary grid resulted in a bias towards superficial layers that was similar in V1, V2 and V3. By contrast, in our study, motion induction inside a surface yielded more superficial activity in V1, but deeper activity in V2 and V3. So, whereas attention likely contributed to both our own data as well as those of Lawrence et al., 2018, it is plausible that the different results between the two studies are caused by the different manners in which the target of attention were defined in the two studies, namely a luminance defined grid in one case, and an induced motion feature in the other case.

Nevertheless, we acknowledge that the question what the precise balance is of attentional contributions versus motion induction is interesting. We have considered the reviewer’s suggestions, and reasoned as follows: If we impose in a few conditions an increase in the strictness of the fixation task, and we observed that the magnitude of the increased signal in the right hemisphere stimulus representation (compared to control conditions) decreases, this can represent both a decrease in attention or attentional capture, and a decrease of the assignment of motion to the surface. So, a parametric decrease in the signal with increasing strictness of the fixation task is not easily interpretable. Alternatively, it could be the case that a differential signal (between experimental and control stimuli) survives, irrespective of the attentional manipulation. This finding in itself is also not interpretable. To show that this signal is specifically related to the motion assignment, and not to other aspects that segregate the figure from the background, one would also have to do a parametric variation of the motion strength and at least one other physical stimulus variable (e.g., the contrast difference between surface and background). In addition, a precise interpretation of the fMRI data would require parallel psychophysical experiments that would have to calibrate double-task paradigms in order to separately titrate effects of attention (to fixation) and motion effects (induced inside the figure surface). In sum, we believe we have plausible arguments against more general exclusively attentional explanations and suggest that the detailed experiments to try to isolate contributions of the motion percept per se from attentional contributions fall outside the scope of the present work.

In the light of the above considerations, we have added a paragraph in the Discussion in which we offer the following interpretation of our data (as well as its limitations):

*“*Attention

Neurophysiological data suggest that the perceptual aspect of a surface is not filled in in the absence of attention (e.g., Poort et al., 2012) and that neural correlates of surface perception are not observed during anesthesia (Lamme et al., 1998). […] However, separating the contributions of attention on the one hand and motion analysis in visual scenes on the other hand to depth-resolved feedback-related fMRI signals is challenging and requires a further series of fMRI and psychophysical experiments outside the scope of the present work.”

[Editors' note: further revisions were suggested prior to acceptance, as described below.]

Revisions for this paper:1) Relationship with attention.The authors have opted not to run an additional control experiment, but rather address the potential role of attention in their findings in Discussion. That is fine in principle, but we do find the new paragraph to be unsatisfying and unclear. Perhaps there may be some confusion about the nature of the concern. We realize that attention may play an intricate (perhaps necessary) role in the cortical computation underlying surface segmentation. But that is not the issue here: the issue is whether the reported effects are (i) due to this computation per se (whether or not some form of attention is involved), or (ii) due to a purely secondary effect of attention being drawn to a segmented surface.This matters for how the results are conceptualized and understood by the broad readership: The first would constitute a specific neural correlate of surface segmentation per se, whereas the second would be a non-specific signature of any form of object perception (compared to no object).We acknowledge that arbitrating between these two scenarios is generally a hard problem in visual neuroscience; yet, there are a few compelling demonstrations in the literature for doing so. We feel that, without using similar manipulations here, there is just now way of knowing, which of these two scenarios accounts for the current results. Future experiments will be needed to pinpoint this.These points, and in particular the two possible scenarios, should be made explicit in the new Discussion paragraph.

We acknowledge that an interpretation of the top-down effect as a secondary effect of attention being drawn to the surface cannot be excluded, and have adjusted the respective paragraph in the Discussion section to clarify this:

“Attention

Although our data do not support that an imbalance in stimulation or attention drove the observed pattern of results, the fixation task we used likely permitted at least a minimal level of attention to the visual field as a whole. […] We cannot fully exclude that our observations reflect a general attentional contribution rather than computations specifically related to the perception of surface motion. These are exciting open questions, and it is encouraging that solving these questions is now within reach of high-field fMRI.”

2) Negative BOLD response due to stimulus with textured background.We remain puzzled by the negative BOLD. Intriguingly, the onset of a whole-field textured background (Figure 6—figure supplement 3) elicits an fMRI response that is comparable in positive amplitude to the negative effects within the stimulus region (Figure 4 and 7). This might indicate that in the main experiment, the textured background causes ongoing positive responses throughout the run, presumably maintained by ongoing micro saccades and fixational instability. Then, the presentation onset of a stimulus has a two-fold effect; it decreases the ongoing response to this background (because it disappears) while also playing the role of an activating stimulus very locally (dark patches in the texture are brighter, whereas light patches are darker, stimulating both on and off visual channels temporarily). In this scenario, the negative response shown in Figure 4 should be seen as a combination of a positive and a negative component, that summed together produce the appearance of a delayed negative response.If the negative BOLD response is indeed a composite of a positive and negative BOLD response, it would be interesting to see how the laminar effects can perhaps decompose this composite. A laminar analysis conducted on the first and later parts of the response separately may be highly insightful here. We understand that SNR may be insufficient to conclusively accommodate this analysis.This idea (possibly others) deserves unpacking, over and above the brief hint towards this possibility the authors mention in subsection “Background dependence of the negative response”. The fact that the authors have seen similar responses in other contexts doesn't necessarily speak to mechanism in this case.

To shed more light on the issue of the negative BOLD response, we included two additional supplementary figures, presenting a simulation on a possible explanation for the delayed negative response, and a laminar analysis regarding the early and late phase of the negative BOLD response. We reference the new supplementary figure at the appropriate location in the Discussion:

“[…] The results lend further support to the hypothesis that the negative signal in the main experiment was driven by a relative lack of activation in response to a uniform stimulus compared to a texture surface. A simulation lends further support to this interpretation (Figure 4—figure supplement 4). Thus, we suggest that the negative surface response resulted primarily from an elevated baseline activity due to the texture background.”

The new simulation suggests that an elevated baseline (caused, for example, by ongoing activation due to the texture background and saccadic eye movements) and a small positive response (caused by the grey stimulus surface) could indeed account for the observed delayed negative response. The signal amplitudes chosen for the simulation are based on empirical data from the additional control experiments, and are therefore plausible.

Regarding the cortical depth profiles of the early and late response phases, we analyzed the early and late phase of the responses, as the reviewer suggested and found no evidence for a qualitatively different laminar profile. A reference to this new figure was added:

“Separate cortical depth profiles of the early and late response phases show no evidence for temporal differences in the laminar activation profile (Figure 4—figure supplement 5).”

3) Relationship/discrepancy with previous animal neurophysiology studies:The authors state in rebuttal: "Finally, electrophysiological experiments typically measure responses over a time window of only a few hundred milliseconds after stimulus onset (300 ms in case of Self et al., 2013). However, the initial and the sustained response to a surface stimulus can have different laminar profiles in V1 (Maier et al., 2011)."If anything, this makes the discrepancy more puzzling: sustained responses, which the authors imply are more strongly reflected in the BOLD response, are predominant in the deep, not the middle and superficial layers (Maier et al., 2011; Self et al., 2013). This part of the Discussion needs revision.

We have revised the respective section, and have removed the unclear argument. Several possible explanations for the discrepancy with electrophysiological studies remain, but we can obviously only speculate on this:

“Relationship to electrophysiological findings

Self et al. (Self et al., 2013) studied the laminar profile of figure-ground segregation in monkey V1. They observed neuronal activity related to feedforward, horizontal, and feedback mechanisms, that are thought to reflect processing of stimulus texture, borders, and figure-ground segregation, respectively. […] Furthermore, beyond the differences in signal measured, stimulus, and experimental design, it is difficult to use the few 100ms typically measured post-stimulus onset in neurophysiological experiments as a predictor for fMRI activity measured 10s and more after stimulus onset.”

4) Possibility of visual cortex effects being the result of top-down feedback to the LGN being fed forward to V1.The authors seem to simply dismiss this possibility, which we find puzzling; a previous study (Akin et al) showed that the experimental modulation used here affects LGN, and depth dependent profiles as reported here in Figure 3, i.e. mostly middle and superficial activations, seems to be very plausibly in line with the effect arising from the LGN, especially when taking the deep-to-superficial draining vein effects into account. Of course, the authors performed a deconvolution to address drain, but we cannot assume that this got rid of any draining vein effects completely. Again, this alternative explanation is worth discussing, rather than simply dismissing it.

We agree that the LGN might have played a role, and have adjusted the Discussion section accordingly:

“An additional contribution to the depth-pattern of activity observed in extrastriate areas may have originated from the pulvinar, the LGN, and possibly other subcortical structures (Standage and Benevento, 1983; Trojanowski and Jacobson, 1977). […] Moreover, an involvement of the LGN in the perception of illusory motion has been observed by Akin et al., 2014, using an experimental design very similar to ours. As the V1 cortical depth profile we observed suggests similar levels of activity in mid-level to superficial layers (Figure 3), it is possible that a feedback signal assigning motion to the grey surface re-entered the LGN, was fed-forward to V1, and from V1 to V2 and V3. In summary, both cortical and subcortical sources of re-entrant feedback in lower-level visual areas may have contributed to the observed depth-resolved responses (see Figure 8).”